# The Empirical Impact of Neural Parameter Symmetries, or Lack Thereof

**Derek Lim**[*]
MIT CSAIL
dereklim@mit.edu

**Theo (Moe) Putterman**[*]
UC Berkeley
moeputterman@berkeley.edu

**Robin Walters**
Northeastern University

**Haggai Maron**
Technion, NVIDIA

**Stefanie Jegelka**
TU Munich, MIT

## Abstract

Many algorithms and observed phenomena in deep learning appear to be affected by parameter symmetries — transformations of neural network parameters that do not change the underlying neural network function. These include linear mode connectivity, model merging, Bayesian neural network inference, metanetworks, and several other characteristics of optimization or loss-landscapes. However, theoretical analysis of the relationship between parameter space symmetries and these phenomena is difficult. In this work, we empirically investigate the impact of neural parameter symmetries by introducing new neural network architectures that have reduced parameter space symmetries. We develop two methods, with some provable guarantees, of modifying standard neural networks to reduce parameter space symmetries. With these new methods, we conduct a comprehensive experimental study consisting of multiple tasks aimed at assessing the effect of removing parameter symmetries. Our experiments reveal several interesting observations on the empirical impact of parameter symmetries; for instance, we observe linear mode connectivity between our networks without alignment of weight spaces, and we find that our networks allow for faster and more effective Bayesian neural network training. Our code is available at https://github.com/cptq/asymmetric-networks.

## 1  Introduction

Neural networks have found profound empirical success, but have many associated behaviors and phenomena that are difficult to understand. One important property of neural networks is that they generally have many *parameter space symmetries* — for any set of parameters, there are typically many other choices of parameters that correspond to the same exact neural network function [24]. For instance, permutations of hidden neurons in a multi-layer perceptron (MLP) induce permutations of weights that leave the overall input-output relationship unchanged. These parameter symmetries are a type of (not-necessarily detrimental) redundancy in the parameterization of neural networks, that adds much non-Euclidean structure to parameter space.

Parameter space symmetries appear to influence several phenomena observed in neural networks. For example, when linearly interpolating between the parameters of two independently trained networks with the same architecture, the intermediate networks typically perform poorly [59, 13]. However, if we first align the two networks via a permutation of parameters that does not affect the network function, then the intermediate networks can perform just as well as the unmerged networks [59, 1]. In some sense, this suggests that neural network loss landscapes are more convex

---

[*]Equal contribution

38th Conference on Neural Information Processing Systems (NeurIPS 2024).

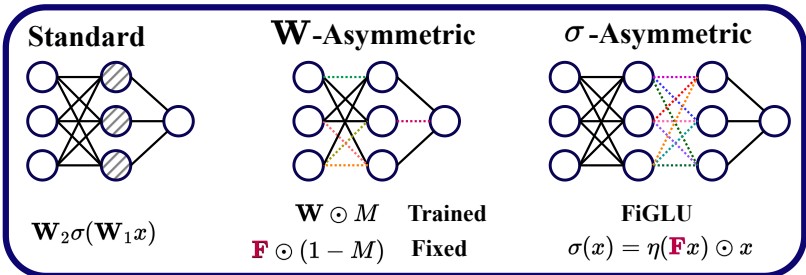

Figure 1: (Left) Standard MLP. The hidden nodes (grey hatches) can be freely permuted, which induces permutation parameter symmetries. Black edges denote trainable parameters. (Middle) Our $\mathbf{W}$-Asymmetric MLP, which fixes certain weights to be constant and untrainable (colored dashed lines) to break parameter symmetries. (Right) Our $\sigma$-Asymmetric MLP, which uses our FiGLU nonlinearity involving a fixed matrix $\mathbf{F}$ (colored dashed lines) to break parameter symmetries.

or well-behaved after removing permutation symmetries. Other areas that parameter symmetries play a role in include interpretability of neurons [20], optimization [48, 84, 80], model merging [60], learned equivariance [7], Bayesian deep learning [34], loss landscape geometry [54], processing neural network weights as input data using metanetworks [39], and generalization measures [49, 11].

To rigorously study the effect of parameter symmetries, we study the effect of removing them. In particular, we introduce two ways of modifying neural network architectures to remove parameter space symmetries (see Figure 1):

(1) $\mathbf{W}$-Asymmetric networks fix certain elements of each linear map to break symmetries in the computation graph.
(2) $\sigma$-Asymmetric networks use a new nonlinearity (FiGLU) that does not act elementwise, and hence does not induce symmetries such as permutations.

These two approaches are inspired by previous work, which shows that both symmetries of computation graphs [39] and equivariances of nonlinearities [20] induce parameter symmetries in standard neural networks. We theoretically prove that both of our approaches remove parameter symmetries under certain conditions. Our Asymmetric networks are similar structurally to standard networks and can be trained with standard backpropagation and first-order optimization algorithms like Adam. Thus, they are a reasonable "counterfactual" system for studying neural networks that are similar to standard neural networks, but that do not have as many parameter symmetries.

With our Asymmetric networks, we run a suite of experiments to study the effects of removing parameter symmetries on several base architectures, including MLPs, ResNets, and graph neural networks. We investigate linear mode connectivity, Bayesian deep learning, metanetworks, and monotonic linear interpolation. Through the lenses of linear mode connectivity and monotonic linear interpolation, we see that the loss landscapes of our Asymmetric networks are remarkably more well-behaved and closer to convex than the loss landscapes of standard neural networks. When using our Asymmetric networks as the base model in a Bayesian neural network, we find faster training and better performance than using standard neural networks that have many parameter symmetries. When using metanetworks to predict properties such as test accuracy of an input neural network, we see that all tested metanetworks more accurately predict the accuracy of Asymmetric networks than standard networks. Overall, our Asymmetric networks provide valuable insights for empirical study and hold promise for advancing our understanding of the impact of neural parameter symmetries.

## 2  Background and Definitions

Let $\Theta$ be the space of parameters of a fixed neural network architecture. For any choice of parameters $\theta \in \Theta$, we have a neural network function $f_\theta : \mathcal{X} \to \mathcal{Y}$ from an input space $\mathcal{X}$ to an output space $\mathcal{Y}$. We call a function $\phi : \Theta \to \Theta$ a *parameter space symmetry* if $f_\theta(x) = f_{\phi(\theta)}(x)$ for all inputs $x$ and parameters $\theta \in \Theta$ (i.e. if $f_\theta$ and $f_{\phi(\theta)}$ are always the same function).

For instance, consider a two-layer MLP with no biases, parameterized by matrices $\theta = (\mathbf{W}_2, \mathbf{W}_1)$ with an elementwise nonlinearity $\sigma$. Then $f_\theta(x) = \mathbf{W}_2 \sigma(\mathbf{W}_1 x)$. Let $P$ be a permutation matrix,

and let $\phi(\theta) = (\mathbf{W}_2 P^\top, P\mathbf{W}_1)$. Then for any input $x$,

$$f_{\phi(\theta)}(x) = \mathbf{W}_2 P^\top \sigma(P\mathbf{W}_1 x) = \mathbf{W}_2 P^\top P \sigma(\mathbf{W}_1 x) = \mathbf{W}_2 \sigma(\mathbf{W}_1 x) = f_\theta(x), \qquad (1)$$

so $\phi$ is a parameter space symmetry. A key step is the second equality, which holds because $P\sigma(x) = \sigma(Px)$: any elementwise nonlinearity $\sigma$ is permutation equivariant. Any other equivariance of $\sigma$ also induces a parameter symmetry; for instance, if $\sigma(x) = \max(0, x)$ is the ReLU function, then $\alpha\sigma(x) = \sigma(\alpha x)$ for any $\alpha > 0$, so there is a positive-scaling-based parameter symmetry [49, 11, 20].

## 3   Related Work

**Characterizing parameter space symmetries.** While many works spanning several decades have noted specific parameter space symmetries in neural networks [24, 61], some works take a more systematic approach to deriving parameter space symmetries. Godfrey et al. [20] characterize all global linear symmetries induced by the nonlinearity for two-layer multi-layer perceptrons with pointwise nonlinearities. Zhao et al. [78] study several types of symmetries, and derive nonlinear, data-dependent parameter space symmetries. Lim et al. [39] show that graph automorphisms of the computation graph of a neural network induce permutation parameter symmetries, which captures hidden neuron permutations in MLPs and hidden channel permutations in CNNs.

**Constraints and post-processing to break parameter space symmetries.** Several works develop methods for constraining or post-processing the weights of a single neural network to remove ambiguities from parameter space symmetries. This includes methods to remove scaling symmetries induced by normalization layers or positively-homogeneous nonlinearities such as ReLU [6, 55, 54, 36], methods to remove permutation symmetries [55, 54, 71, 36], and methods to remove sign symmetries induced by odd activation functions [71].

Unlike these previous works, we develop neural network architectures that have reduced parameter space symmetries. Our models are optimized using standard unconstrained gradient-descent based methods like Adam. Hence, our networks do not require any non-standard optimization algorithms such as manifold optimization or projected gradient descent [6, 55], nor do they require post-training-processing to remove symmetries or special care during analysis of parameters (such as geodesic interpolation in a Riemannian weight space [54]). These methods based on constraining weights or post-processing have significantly different optimization and loss landscape properties (for instance, linear interpolation is not even well-defined on general nonlinear parameter manifolds), so they are less suitable than our Asymmetric networks for studying phenomena that may generalize to standard neural networks.

**Aligning multiple networks for relative invariance to parameter symmetries.** One way to reduce the impact of parameter symmetries in certain settings, especially for model merging, is to align the parameters of one network to another. Several methods have been proposed for choosing permutations that align the parameters of two neural networks of the same architecture, via efficient heuristics or learned methods [4, 69, 63, 13, 1, 53, 47, 67]. Other approaches relax the exact permutation-parameter-symmetry constraint or do additional postprocessing to achieve effective merging of models in parameter space [59, 28, 29, 60, 56]. As our Asymmetric networks have removed parameter symmetries, they can often be successfully merged and linearly interpolated between without any alignment.

## 4   Asymmetric Networks

We develop two methods of parameterizing neural network architectures without parameter symmetries, both of which are justified by theoretical results. We can prove that our $\sigma$-Asym networks have permutation and scale symmetries removed, and that our $\mathbf{W}$-Asym networks have permutation symmetries removed. Although we have not formally proven that $\mathbf{W}$-Asym networks have scale symmetries removed, we believe that they do (intuitively, the fixed weights fix a scale).

We first focus on the case of fully-connected MLPs with no biases, which take the form $f_\theta(x) = \mathbf{W}_L \sigma(\mathbf{W}_{L-1} \cdots \sigma(\mathbf{W}_1 x))$ for weights $\theta = (\mathbf{W}_L, \ldots, \mathbf{W}_1)$ and nonlinearity $\sigma$. Then in Section 4.3, we discuss how we use these approaches to remove parameter symmetries in other architectures (e.g. CNNs, GNNs) as well.

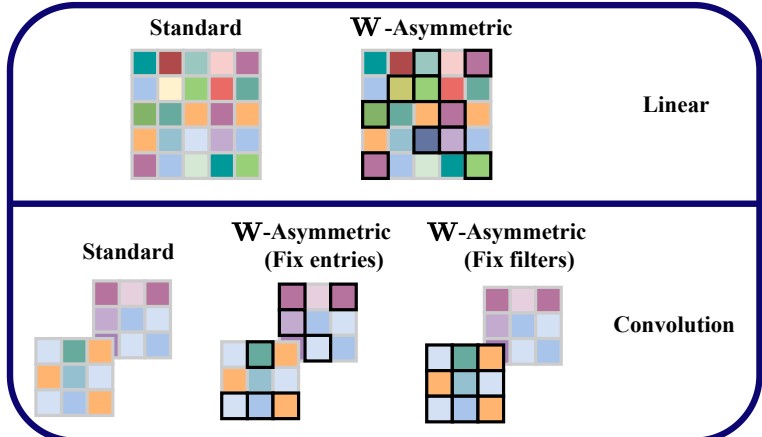

Figure 2: Depiction of our **W**-Asymmetric approach to removing parameter symmetries. Entries with a black outline are untrained. Note that the **W**-Asym linear map has 2 nonzeros per row, the **W**-Asym convolution with fixed entries has 8 fixed entries for its single output channel, and the **W**-Asym convolution with fixed filters has a single input filter fixed. We often use a constant number of fixed entries per row or output channel in our experiments.

## 4.1 Computation Graph Approach (W-Asymmetric Networks)

Our first approach to developing neural networks with greatly reduced parameter space symmetries relies on their computation graph. In particular, we can write a feedforward neural network architecture as a DAG $G = (V, E)$ with neurons as nodes $V$ and connections between them as edges $E$. For a choice of parameters $\theta \in \mathbb{R}^{|E|}$, we get a function $f_\theta$ from input neuron space to output neuron space [19, 49, 39]

Lim et al. [39] showed that neural DAG automorphisms $\phi$, which are graph automorphisms of the DAG $G$ that preserve types of nodes and weight-sharing constraints, induce permutation parameter symmetries $\phi$ that leave the function unchanged: $f_\theta = f_{\phi(\theta)}$. Thus, any feedforward neural network architecture that has no permutation parameter symmetries must necessarily have a computation graph with no nontrivial neural DAG automorphisms.

To modify MLPs so they have no nontrivial neural DAG automorphisms, we mask edges in the computation graph, by setting certain edge weights to constant values that are not updated during training. For an MLP, we can do this by enforcing that every linear layer $T : \mathbb{R}^{d_1} \to \mathbb{R}^{d_2}$ takes the form of a matrix $\mathbf{W} \in \mathbb{R}^{d_2 \times d_1}$, where each row has a unique pattern of untrained weights. To achieve this, define a mask $M \in \{0, 1\}^{d_2 \times d_1}$ such that $\mathbf{W}_{ij}$ is a trainable parameter if and only if $M_{ij} = 1$, and the rows of $M$ are pairwise distinct binary vectors in $\{0, 1\}^{d_1}$. We call any neural network with linear maps masked as such a **W**-Asymmetric neural network. In Appendix B.1, we show that masking these entries so that they are not trained is sufficient to remove all nontrivial neural DAG automorphisms.

**Theorem 1.** *If each mask matrix $M$ has unique nonzero rows, then* **W**-*Asymmetric MLPs with fixed entries set to zero have no nontrivial neural DAG automorphisms.*

In practice, we generate a binary mask $M$ by randomly selecting a subset of $n_{\text{fix}}$ fixed elements for each row. For the fixed entries, we sample them from a normal distribution $\mathcal{N}(0, \kappa I)$ with standard deviation $\kappa > 0$ that is a hyperparameter that we tune. Our asymmetric linear layer can be written as

$$\mathbf{W}' = M \odot \mathbf{W} + (1 - M) \odot \mathbf{F}, \tag{2}$$

where $\mathbf{W} \in \mathbb{R}^{d_2 \times d_1}$ is a matrix of trainable parameters, and $\mathbf{F} \in \mathbb{R}^{d_2 \times d_1}$ is a matrix of fixed elements, sampled from $\mathcal{N}(0, \kappa I)$. The only trainable parameters are the unmasked entries of $M \odot \mathbf{W}$, of which there are $d_2 \cdot (d_1 - n_{\text{fix}})$. We empirically find that having $\kappa$ be significantly larger than the standard deviation of typical initializations for weight matrices (e.g. $\kappa = 1$ while the trained coefficients have standard deviation about $1/\sqrt{1000}$) is important for breaking parameter symmetries.

## 4.2 Nonlinearity Approach ($\sigma$-Asymmetric Networks)

Another approach for removing parameter symmetries is to change the nonlinearity. As studied by Godfrey et al. [20], equivariances of the nonlinearity induce parameter symmetries in MLPs with elementwise nonlinearities. Recall that an elementwise nonlinearity acts by using the same function on each coordinate of the input; $\sigma : \mathbb{R}^d \to \mathbb{R}^d$ is elementwise if it takes the form $\sigma(x) = (\sigma_1(x_1), \ldots, \sigma_1(x_d))$ for some real function $\sigma_1 : \mathbb{R} \to \mathbb{R}$. Any elementwise nonlinearity is permutation equivariant, and hence induces a permutation parameter symmetry.

Thus, in contrast to most neural network architectures, for Asymmetric networks we must use a nonlinearity that does not act elementwise. Likewise, the nonlinearity cannot have any linear symmetry itself, since if $\sigma \circ A = B \circ \sigma$ for $A, B \in GL(d)$, then for a two-layer network:

$$\mathbf{W}_2 \circ \sigma \circ \mathbf{W}_1 = \mathbf{W}_2 B^{-1} B \circ \sigma \circ \mathbf{W}_1 = \mathbf{W}_2 B^{-1} \circ \sigma \circ A \mathbf{W}_1. \tag{3}$$

So $(\mathbf{W}_2, \mathbf{W}_1)$ and $(\mathbf{W}_2 B^{-1}, A\mathbf{W}_1)$ give the same neural network function. Thus, in order to define a model class without parameter symmetries, it is necessary for $\sigma$ to have *no linear equivariances*, i.e. we desire that if $\sigma \circ A = B \circ \sigma$ for $A, B \in GL(d)$, then $A = B = I$. For two-layer MLPs with square invertible weights, this is in fact sufficient to remove all parameter symmetries: we prove this in Appendix B.2.

**Proposition 1.** *Let the parameter space $\Theta$ be all pairs of square invertible matrices $\theta = (\mathbf{W}_2, \mathbf{W}_1)$ for $\mathbf{W}_2, \mathbf{W}_1 \in GL(d)$, and let $f_\theta(x) = \mathbf{W}_2 \sigma(\mathbf{W}_1 x)$. If $\sigma$ has no linear equivariances, then $f_{\theta_1} = f_{\theta_2}$ if and only if $\theta_1 = \theta_2$. In other words, there are no nontrivial parameter space symmetries.*

### 4.2.1 FiGLU: the Fixed Gated Linear Unit Nonlinearity

Motivated by Proposition 1, we define a non-elementwise nonlinearity that does not have the equivariances of standard nonlinearities. Letting $\eta$ be the sigmoid function $\eta(x) = \frac{1}{1+e^{-x}}$, we define our nonlinearity as

$$\sigma(x) = \eta(\mathbf{F}x) \odot x, \tag{4}$$

for a randomly sampled, untrained matrix $\mathbf{F}$. Similarly to $\mathbf{W}$-Asym nets, we sample $\mathbf{F}$ as an i.i.d Gaussian matrix with variance that we tune. This nonlinearity is similar to Swish / SiLU [57, 26] with an additional matrix $\mathbf{F}$ to mix feature dimensions (to break permutation equivariance), and it is also similar to a gated linear unit (GLU) with no trainable parameters [9]. Thus, we call our nonlinearity FiGLU: the Fixed Gated Linear Unit.

In Appendix B.2.1, we prove that FiGLU does not have permutation equivariances or diagonal equivariances, which are the only equivariances for most elementwise nonlinearities [20].

**Proposition 2.** *With probability $1$ over the sampling of $\mathbf{F}$, FiGLU has no permutation equivariances or diagonal equivariances.*

We call any network with our symmetry-breaking FiGLU nonlinearity a $\sigma$-Asymmetric Network.

## 4.3 Extension to Other Architectures

The graph-based approach ($\mathbf{W}$-Asymmetric Networks) works naturally for neural network architectures with "channel" dimensions, such as convolutional neural networks (CNNs), graph neural networks (GNNs) [22], Transformers [66], and equivariant neural networks based on equivariant linear maps [16]. In these types of networks, permutations of entire channels induce permutation parameter symmetries [39]. For such networks, we thus mask entire connections between channels, e.g. entire filters in CNNs. For CNNs, we also experiment with randomly masking some number of entries in each filter (instead of masking entire filters), and find that this also works well in removing parameter symmetries. For neural networks with linear layers that include bias terms, we do not modify the biases in any way, as they do not introduce new computation graph automorphisms [39].

The nonlinearity-based approach ($\sigma$-Asymmetric Networks) can be straightforwardly applied to many general architectures as well. Though, the fixed matrix $\mathbf{F}$ may have to be changed to a structured linear map; for instance, in CNNs we take $\mathbf{F}$ to be a 1D convolution.

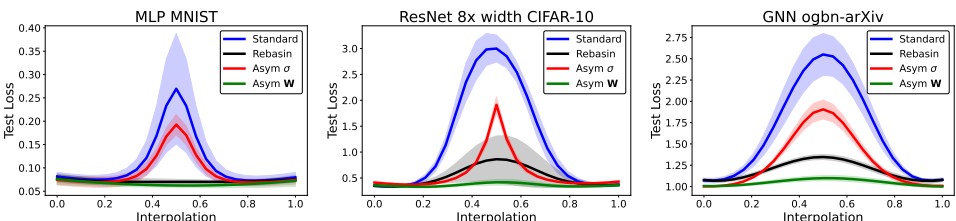

Figure 3: Linear mode connectivity: test loss curves along linear interpolations between trained networks. (Left) MLP on MNIST. (Middle) ResNet with $8\times$ width on CIFAR-10. (Right) GNN on ogbn-arXiv. $\mathbf{W}$-Asymmetric networks interpolate the best, followed by networks aligned with Git-Rebasin, then $\sigma$-Asymmetric networks, and finally standard networks.

## 4.4 Universal Approximation

Our two approaches remove parameter symmetries from standard neural networks, but still intuitively retain much of the structure of standard networks. One important property of widely-used neural network architectures is universal approximation — for any target function of a certain type, there exists a neural network of the given architecture that approximates the target to an arbitrary accuracy [8, 25, 42, 75]. In Appendix B.3, we show that $\mathbf{W}$-Asymmetric MLPs retain this property:

**Theorem 2** (Informal). *For $n_{\text{fix}} \in o(n^{\frac{1}{4}})$, where $n$ is the hidden dimension, $\mathbf{W}$-Asymmetric MLPs are Universal Approximators with probability $1$ over the choice of hardwired entries.*

We have not been able to prove a similar result for $\sigma$-Asymmetric networks. Classical universal approximation results for standard neural networks do not apply to $\sigma$-Asym nets, as they tend to assume elementwise nonlinearities.

## 5 Experiments

### 5.1 Linear Mode Connectivity without Permutation Alignment

**Background.** Under certain conditions, neural networks have been found to exhibit linear mode connectivity, which is when all networks on the line segment in parameter space between two well-performing trained networks are also well-performing. Starting with Frankle et al. [18], who coined the term and provided the first in-depth analysis, many works have studied this phenomenon [44, 13, 76, 14]. When the two networks are randomly initialized and trained independently, linear mode connectivity generally does not hold [13, 1]. However, if one of the two networks is permuted with a parameter symmetry that does not change its function, but that aligns its parameters with the other network, then linear mode connectivity empirically and theoretically holds for many more model / task combinations [13, 1, 79, 14]. In fact, Entezari et al. [13] conjectures that if all permutation symmetries are accounted for, then linear mode connectivity generally holds. Since our Asymmetric networks remove parameter space symmetries, we may expect linear mode connectivity to hold, without any post-processing or alignment step.

**Hypothesis.** Asymmetric networks are more linearly mode connected than standard networks, and do not require post-processing or alignment of pairs of networks before merging.

**Experimental Setup.** We consider several networks and tasks: MLPs on MNIST, ResNets [23] on CIFAR-10, and Graph Neural Networks [73] on ogbn-arXiv [27]. For each architecture and task, we compute the midpoint test loss barrier: $L(\frac{1}{2}\theta_1 + \frac{1}{2}\theta_2) - \frac{1}{2}(L(\theta_1) + L(\theta_2))$. This measures how much worse the interpolated network with parameters $\frac{1}{2}\theta_1 + \frac{1}{2}\theta_2$ is than the original networks with parameters $\theta_1$ and $\theta_2$. We measure this barrier for standard networks, pairs of networks aligned with Git-Rebasin [1], and networks with our two approaches ($\sigma$-Asym and $\mathbf{W}$-Asym) applied. To be clear, whenever we interpolate between the weights of two Asymmetric networks, they have the same exact fixed weights $\mathbf{F}$ (for both $\mathbf{W}$-Asym and $\sigma$-Asym) and the same exact mask $M$ (for $\mathbf{W}$-Asym).

**Results.** Figure 3 plots interpolation curves and Table 1 displays midpoint test loss barriers of various methods. Our $\sigma$-Asymmetric approach lowers the test loss barrier compared to standard networks, but falls short of the alignment approach of Git-Rebasin. On the other hand, our $\mathbf{W}$-Asymmetric

approach achieves strong (and sometimes perfect) interpolation, and interpolates better than standard networks aligned via Git-ReBasin. This may be caused by failure of the Git-ReBasin approaches to find the optimal permutations, importance of other parameter symmetries besides layer-wise permutations, or other properties of $\mathbf{W}$-Asymmetric networks.

Table 1: Test loss interpolation barriers at midpoint: $L(\frac{1}{2}\theta_1 + \frac{1}{2}\theta_2) - \frac{1}{2}(L(\theta_1) + L(\theta_2))$ . We use different methods of breaking symmetries in each column; from left to right: no symmetry breaking, Git-Rebasin [1], our $\sigma$-Asym approach, and our $\mathbf{W}$-Asym approach. We report mean and standard deviation of the barrier across at least 5 pairs of networks, and bold lowest barriers.

|  | Standard | Git-ReBasin | $\sigma$-Asym (ours) | $\mathbf{W}$-Asym (ours) |
|---|---|---|---|---|
| MLP (MNIST) | $0.188 \pm .12$ | $-.006 \pm .00$ | $0.117 \pm .01$ | $\mathbf{-0.012} \pm .00$ |
| ResNet (CIFAR-10) | $3.287 \pm .32$ | $2.041 \pm .21$ | $2.521 \pm .46$ | $\mathbf{0.934} \pm .72$ |
| ResNet 8x width (CIFAR-10) | $2.640 \pm .24$ | $0.509 \pm .45$ | $1.492 \pm .15$ | $\mathbf{0.031} \pm .05$ |
| GNN (ogbn-arXiv) | $1.475 \pm .24$ | $0.269 \pm .02$ | $0.901 \pm .11$ | $\mathbf{0.095} \pm .03$ |

## 5.2 Bayesian Neural Networks

**Background.** Bayesian deep learning is a promising approach to improve several deficits of mainstream deep learning methods, such as uncertainty quantification and integration of priors [30, 51]. However, parameter symmetries are problematic in Bayesian neural networks, as they are a major source of statistical nonidentifiability [30]. Parameter symmetries introduce modes in the posterior $p(\theta|\mathcal{D})$ that make the posterior harder to approximate [2, 34, 72], sample from [50, 71], and otherwise analyze [36]. For instance, one common technique for training Bayesian neural networks is variational inference via fitting a Gaussian distribution to the true posterior $p(\theta|\mathcal{D})$. This approach suffers because the Gaussian distribution has only one mode, whereas the true posterior has at least one mode for every parameter symmetry. As such, some approaches treat kernel matrices are random variables, which has less symmetries than treating features or weights as random variables, and allows better approximation by unimodal posteriors [74, 43]. Instead, we consider traditional, commonly-used Bayesian deep learning techniques applied on the features or weights of our Asymmetric networks.

Table 2: Bayesian neural network results. Reported loss is the negative log likelihood loss. All results (except for last column) are after 50 epochs of training. $\mathbf{W}$-Asymmetric networks tend to improve over their standard counterparts, especially early in training. 16-layer MLPs fail to train, but 16-layer $\mathbf{W}$-Asymmetric MLPs successfully train. Standard or Asymmetric networks better than their counterpart by a standard deviation are bolded.

|  | Model | Train Loss ↓ | Test Loss ↓ | ECE ↓ | Test Acc ↑ | Test Acc (25 Epochs) ↑ |
|---|---|---|---|---|---|---|
| CIFAR-10 | MLP-8 | $1.34 \pm .00$ | $1.24 \pm .01$ | $.039 \pm .009$ | $56.37 \pm .31$ | $52.87 \pm 0.2$ |
|  | $\mathbf{W}$-Asym MLP-8 | $\mathbf{1.31} \pm .01$ | $\mathbf{1.22} \pm .01$ | $.042 \pm .009$ | $\mathbf{57.08} \pm .50$ | $\mathbf{54.15} \pm 0.2$ |
|  | MLP-16 | $2.29 \pm .02$ | $2.28 \pm .03$ | $.026 \pm .017$ | $13.54 \pm 2.0$ | $13.34 \pm 2.7$ |
|  | $\mathbf{W}$-Asym MLP-16 | $\mathbf{1.39} \pm .01$ | $\mathbf{1.27} \pm .01$ | $.045 \pm .009$ | $\mathbf{55.16} \pm .44$ | $\mathbf{51.42} \pm 0.3$ |
| CIFAR-10 | ResNet20 | $\mathbf{.596} \pm .01$ | $.535 \pm .03$ | $.045 \pm .007$ | $81.98 \pm 1.2$ | $72.37 \pm 1.0$ |
|  | $\mathbf{W}$-Asym ResNet20 | $.600 \pm .02$ | $.535 \pm .01$ | $.044 \pm .004$ | $81.94 \pm 0.6$ | $73.64 \pm 1.5$ |
|  | ResNet110 | $.803 \pm .08$ | $.706 \pm .08$ | $.052 \pm .007$ | $75.71 \pm 2.8$ | $59.85 \pm 3.9$ |
|  | $\mathbf{W}$-Asym ResNet110 | $\mathbf{.745} \pm .07$ | $\mathbf{.658} \pm .06$ | $.049 \pm .004$ | $77.40 \pm 2.4$ | $\mathbf{63.20} \pm 3.0$ |
| CIFAR-100 | ResNet20 (BN) | $1.68 \pm .03$ | $1.57 \pm .02$ | $.078 \pm .004$ | $56.83 \pm .62$ | $46.80 \pm 0.9$ |
|  | $\mathbf{W}$-Asym ResNet20 (BN) | $\mathbf{1.62} \pm .02$ | $\mathbf{1.50} \pm .03$ | $.076 \pm .006$ | $\mathbf{58.40} \pm .62$ | $\mathbf{49.29} \pm 0.4$ |
|  | ResNet20 (LN) | $1.97 \pm .02$ | $1.88 \pm .02$ | $.090 \pm .007$ | $50.02 \pm .54$ | $37.24 \pm 1.1$ |
|  | $\mathbf{W}$-Asym ResNet20 (LN) | $\mathbf{1.91} \pm .03$ | $\mathbf{1.82} \pm .02$ | $.086 \pm .006$ | $\mathbf{51.20} \pm .47$ | $\mathbf{39.03} \pm 1.0$ |

**Hypothesis.** Using Asymmetric networks as the base model improves Bayesian neural networks, as the posterior will have less modes.

**Experimental setup.** We train Standard Bayesian and Asymmetric Bayesian Networks for image classification using variational inference. We use the method of [64] for variational inference, which fits a Gaussian approximate posterior with a diagonal plus low-rank covariance. We train 10 instances of each model and then report train loss, test loss, test accuracy, and Expected Calibration Error (ECE) [45], which is a measure of calibration.

**Results.** See training curves in Figure 4, and quantiative results in Table 2. Using $\mathbf{W}$-Asymmetric networks as a base for Bayesian deep learning improves training speed and convergence. Most

strikingly, Bayesian MLPs of depth 16 cannot train at all, while **W**-Asymmetric Bayesian MLPs train well. In general, the **W**-Asymmetric approach improves training and test accuracy across the several models (MLPs, ResNets of varying sizes, and ResNets with either batch norm or layer norm).

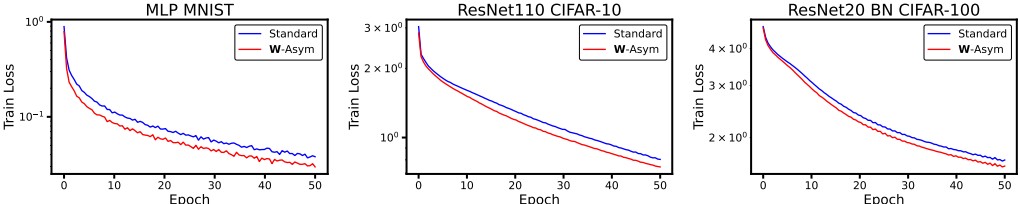

Figure 4: Bayesian neural network training loss over time for depth 8 MLPs on MNIST (left), ResNet110 on CIFAR-10 (middle), and ResNet20 with BatchNorm on CIFAR-100 (right). **W**-Asymmetric networks train more quickly, and achieve lower training loss.

## 5.3 Metanetworks

Table 3: Metanetwork performance for predicting the test accuracy of small ResNets and our **W**-Asym ResNets. Each row is a different metanetwork. Reported are $R^2$ and Kendall $\tau$ on the test set — higher is better.

|  | ResNet | | **W**-Asym ResNet | |
|---|---|---|---|---|
|  | $R^2$ | $\tau$ | $R^2$ | $\tau$ |
| MLP | $.330 \pm .04$ | $.389 \pm .03$ | $\mathbf{.594} \pm .12$ | $\mathbf{.864} \pm .01$ |
| DMC [12] | $.950 \pm .01$ | $.787 \pm .02$ | $\mathbf{.967} \pm .01$ | $\mathbf{.911} \pm .01$ |
| DeepSets [77] | $.855 \pm .01$ | $.617 \pm .03$ | $\mathbf{.936} \pm .00$ | $\mathbf{.858} \pm .00$ |
| StatNN [65] | $.976 \pm .00$ | $.866 \pm .00$ | $\mathbf{.978} \pm .00$ | $\mathbf{.935} \pm .01$ |

**Background.** Metanetworks [39] — also referred to as deep weight-space networks [46, 58], meta-models [35], or neural functionals [81, 82, 83] — are neural networks that take as inputs the parameters of other neural networks. Recent work has found that making metanetworks invariant or equivariant to parameter-space symmetries of the input neural networks can substantially improve metanetwork performance [46, 81, 39, 32].

**Hypothesis.** Asymmetric networks are easier to train metanetworks on because they do not have to explicitly account for symmetries.

**Experimental setup.** We experiment with metanetworks on the task of predicting the CIFAR-10 test accuracy of an input image classifier, which many metanetworks have been tested on [65, 12, 81, 39]. We use metanetworks based on simple MLPs, 1D-CNN metanetworks [12], and metanetworks that are exactly invariant to permutation parameter symmetries: DeepSets [77] and StatNN [65]. We train two separate datasets of 10,000 image classifiers: one dataset of small ResNet models, and one dataset of **W**-Asymmetric ResNet models. More information on the data, metanetworks, and training details are in Appendix F.3.

**Results.** In Table 3, we see that metanetworks are signfinacntly better at predicting the performance of our **W**-Asymmetric ResNets than standard ResNets. Interestingly, simple MLP metanetworks, which view the input parameters as a flattened vector, can predict the test accuracy of Asymmetric Networks quite well, but fail on standard networks. Also, the permutation equivariant metanetworks (DeepSets and StatNN) both improve on **W**-Asym ResNets compared to on ResNets, even though the permutation symmetries of standard ResNets do not affect these metanetworks; thus, it may be possible that other symmetries in standard ResNets (but not Asym-ResNets) harm metanetwork performance, or they may be other factors besides symmetries that improve metanetwork performance for Asym-ResNets.

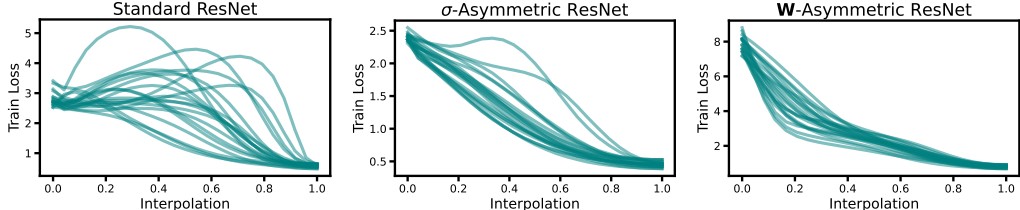

Figure 5: Train loss against interpolation coefficient $\alpha$ for the interpolation $(1 - \alpha)\theta_0 + \alpha\theta_T$ between initial parameters $\theta_0$ and trained parameters $\theta_T$. Trajectories for the 20 $(\theta_0, \theta_T)$ pairs of lowest train loss for each architecture are plotted. The trajectories for Asymmetric ResNets appear significantly more monotonic and convex.

Table 4: Monotonic linear interpolation: properties of linear interpolations between 300 pairs of initialization and trained parameters. Arrows denote behavior that is more similar to convex optimization, e.g. there is a downarrow ($\downarrow$) next to $\Delta$ because convex objectives have nonpositive $\Delta$, while nonconvex can have positive $\Delta$. For both types of Asymmetric networks, all differences from Standard ResNets are statistically significant ($p < .001$) under a two-sided T-test: Asymmetric networks have significantly more monotonic and convex linear interpolations from initialization.

| | $\Delta \downarrow$ | Percent Monotonic $\uparrow$ | Local Convexity $\uparrow$ | Global Convexity $\uparrow$ |
|---|---|---|---|---|
| Standard ResNet | $.079 \pm .109$ | $26.3\%$ | $.548 \pm .139$ | $.823 \pm .229$ |
| $\sigma$-Asym ResNet | $.004 \pm .047$ | $87.3\%$ | $.675 \pm .143$ | $.976 \pm .098$ |
| **W**-Asym ResNet | $-\mathbf{.027} \pm .026$ | $\mathbf{100}\%$ | $\mathbf{.769} \pm .165$ | $\mathbf{1.00} \pm .000$ |

## 5.4 Monotonic Linear Interpolation

**Background.** One common method of studying the loss landscapes of neural networks is by studying the one-dimensional line segment of parameters attained by linear interpolation between parameters at initialization and parameters after training. Many works have observed *monotonic linear interpolation* (MLI), which is when the training loss monotonically decreases along this line segment [21, 17, 41, 68]. Loss landscapes of convex problems have this property as well, so presence of the monotonic linear interpolation property has been used as a rough measure of how well-behaved the loss landscape is. However, with many types of models, tasks, or hyperparameter settings, monotonic linear interpolation does not hold [41, 68], or there is a large plateau where the loss barely changes for much of the line segment [17, 70]; neither of these properties can happen for convex objectives trained to completion. To the best of our knowledge, there has been little work on the role of parameter symmetries — or lack thereof — in monotonic linear interpolation (besides one minor experiment in [41] Appendix C.9). Nonetheless, since removing parameter symmetries substantially improves linear interpolation between trained networks (Section 5.1), one may expect removing parameter symmetries to improve monotonic linear interpolation.

**Hypothesis.** The training loss along the line segment between initialization and trained parameters is more monotonic and convex for Asymmetric networks.

**Experimental setup.** For the learning task, we follow the setup used for creating the dataset of image classifiers in Section 5.3. In particular, we train 300 standard ResNets and 300 **W**-Asymmetric ResNets with varying hyperparameters sampled from the same distributions as used for the dataset of image classifiers (see Appendix Table 13). For each of these networks, we linearly interpolate between its initial parameters $\theta_0$ and its final trained parameters $\theta_T$: $(1 - \alpha)\theta_0 + \alpha\theta_T$ for 25 uniformly spaced values $0 = \alpha_1 < \alpha_2 < \ldots < \alpha_{25} = 1$. To measure monotonicity, we record the maximum increase between adjacent networks $\Delta = \max(L(\alpha_{i+1}) - L(\alpha_i))$, and the percentage of networks that have $\Delta \leq 0$ i.e. the percentage of networks that satisfy monotonic linear interpolation. To measure convexity, we consider a local convexity measure (the proportion of $\alpha_i$ where the centered difference second derivative approximation is nonnegative) and a global convexity measure (the proportion of $\alpha_i$ such that $L(\alpha_i)$ lies below the line segment between the endpoints, i.e. $L(\alpha_i) \leq (1 - \alpha_i)L(0) + \alpha_i L(1)$).

**Results.** Table 4 shows the measures of monotonicity and convexity for standard, $\sigma$-Asymmetric, and **W**-Asymmetric ResNets. Remarkably, every single one of the 300 **W**-Asymmetric ResNets satisfies

monotonic linear interpolation and has a trajectory that lies underneath the line segment between the endpoints. Qualitatively, we can see in Figure 5 that $\mathbf{W}$-Asymmetric ResNets do not have any clear loss barriers from initialization, nor any loss plateaus that indicate nonconvexity. In contrast, the majority of standard ResNets have non-monotonic trajectories, and the monotonic trajectories seem to be more nonconvex. $\sigma$-Asymmetric network trajectories are signficantly more convex and monotonic than standard network trajectories, but there are some non-monotonic or nonconvex trajectories still.

### 5.5 Other Optimization and Loss Landscape Properties

In Appendix A, we note other interesting differences in optimization and loss landscape properties of Asymmetric and standard neural networks. These can be summarized as:

1. Even though Asymmetric networks interpolate much better than standard networks, the parameters of trained Asymmetric networks are often basically the same distance away from each other in weight space as standard networks.
2. Asymmetric networks do not tend to overfit as much: the difference in train performance and test performance can be substantially lower than that of standard networks.
3. Asymmetric networks can take longer to train, especially when choosing hyperparameters that make them more dissimilar to standard networks.

## 6 Discussion

While many properties of Asymmetric networks are in line with our hypotheses and intuition about the impact of removing parameter symmetries, there are many unexpected effects and unanswered questions that are promising to further investigate. For instance, we did not extensively explore Asymmetric networks in the context of model interpretability, generalization measures in weight spaces, or optimization improvements, all of which are known to be influenced to some extent by parameter symmetries. Further studying the properties in Section 5.5, the dependence of behavior on the choices of Asymmetry-inducing hyperparameters, and other design choices in making networks asymmetric could also bring more insights into parameter space symmetries.

Also, it is interesting that our $\sigma$-Asymmetric networks do not appear to break parameter symmetries as well as our $\mathbf{W}$-Asymmetric networks. We have run preliminary empirical tests on several variants of $\sigma$-Asym networks, such as: $\sigma \circ \mathbf{F} \circ \sigma$ as the nonlinearity, sparsifying $\mathbf{F}$, adding instead of multiplying the gate, using cosine instead of sigmoid, squaring instead of using sigmoid, and putting a LayerNorm in the nonlinearity. However, none of these approaches worked well. We believe that such failures may be because $\sigma$-Asymmetry breaks symmetries at the activation / neuron level, whereas $\mathbf{W}$-Asymmetry breaks symmetries in the larger space of weights (for more evidence, see Appendix E, where we show that fixing biases at neurons also fails to effectively remove symmetries). These curiosities provide interesting directions for future work.

All in all, we believe that future theoretical and empirical study of Asymmetric networks could garner many insights into the role of parameter symmetries in deep learning.

#### Acknowledgements

We would like to thank Kwangjun Ahn, Benjamin Banks, Nima Dehmamy, Nikos Karalias, Jinwoo Kim, Marc Law, Hannah Lawrence, Thien Le, Jonathan Lorraine, James Lucas, Behrooz Tahmasebi, and Logan Weber for discussions at various points of this project. DL is supported by an NSF Graduate Fellowship. RW is supported in part by NSF award 2134178. HM is the Robert J. Shillman Fellow, and is supported by the Israel Science Foundation through a personal grant (ISF 264/23) and an equipment grant (ISF 532/23). This research was supported in part by Office of Naval Research grant N00014-20-1-2023 (MURI ML-SCOPE), NSF AI Institute TILOS (NSF CCF-2112665), NSF award 2134108, and the Alexander von Humboldt Foundation.

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

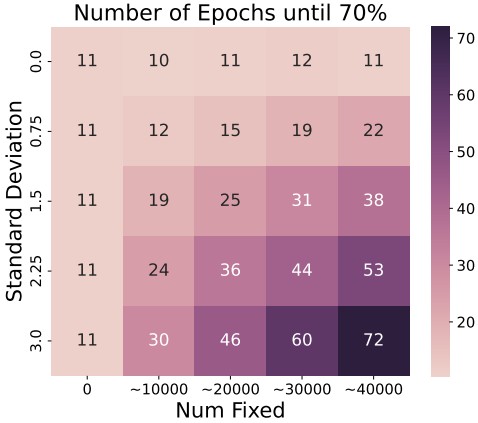

Figure 6: Epochs until reaching 70% training accuracy on CIFAR-10 when varying the hyperparameters of $\mathbf{W}$-Asymmetric ResNets; we vary number of fixed entries $n_{\text{fix}}$ and standard deviation $\kappa$ of the fixed entries $\mathbf{F}$. Entries further to the bottom and right are more asymmetric, while the entries further to the top and left are more like standard networks (the leftmost column are all standard networks). We see that more-asymmetric networks need more time to train.

## A    Additional Observations on Asymmetric Networks

There are several other interesting differences in the optimization and loss landscape properties of Asymmetric and standard neural networks. For one, even though Asymmetric networks generally interpolate significantly better than standard networks, this cannot be seen by measuring distances in parameter space. For instance, in GNN experiments following the setup of Section 5.1, pairs of standard GNNs have a distance per parameter of .000174 on average, whereas $\mathbf{W}$-Asymmetric GNNs have .000159, which is only slightly lower. However, the average test loss barrier is 1.448 for standard GNNs while it is only 0.069 for $\mathbf{W}$-Asymmetric GNNs. Likewise, in our datasets of 10,000 standard and $\mathbf{W}$-Asymmetric ResNets, the average distance per parameter between the weights of trained standard classifiers is .0034, which is actually lower than the distance per parameter of .0051 for $\mathbf{W}$-Asymmetric ResNets (estimated on 20,000 pairs of networks). Thus, although we sometimes imagine well-interpolating pairs of networks to lie in the same local basin of parameter space, $\mathbf{W}$-Asymmetric networks are actually rather far apart in parameter space, but nonetheless have linear segments of low loss between them.

We also find that Asymmetric networks often do not overfit as much as standard networks. For instance, in the GNN setup of Section 5.1, standard GNNs have a max training accuracy of $84.6\%$ on average, with a validation accuracy of $71.6\%$. On the other hand, $\sigma$-Asym GNNs have $70.8\%/70.1\%$ train/validation accuracy, while $\mathbf{W}$-Asym GNNs have $70.7\%/70.06\%$ train/validation accuracy. This difference does not show as much in our datasets of 10,000 standard ResNets and $\mathbf{W}$-Asym ResNets, possibly because of the substantial regularization (data augmentation, weight decay, and label smoothing) used for training (standard gets $74.8\%/73.8\%$ train/test accuracy while $\mathbf{W}$-Asymmetric gets $64.0\%/64.0\%$).

Further, in Figure 6, we see that training speed is slower for $\mathbf{W}$-Asymmetric ResNets when we increase the amount of asymmetry (by increasing the number of fixed entries and the standard deviation of the fixed entries). While standard ResNets take on average 11 epochs to reach 70% training accuracy on CIFAR-10, $\mathbf{W}$-Asymmetric ResNets with the most extreme hyperparameters take up to 72 epochs.

# B  Proofs of Theoretical Results

## B.1  Graph-based approach

Here, we prove that as long as each mask matrix $M$ in our $\mathbf{W}$-Asymmetric MLPs with fixed entries set to zero has unique nonzero rows, then our architecture has no nontrivial neural DAG automorphisms. In practice, we find that setting the standard deviation $\kappa$ of the fixed entries $\mathbf{F}$ to be positive (and in fact orders of magnitude larger than the standard deviation that we typically initialize trainable weights with) is important to achieve properties such as linear mode connectivity that Asymmetric networks have but standard networks do not have. When $\kappa = 0$ (i.e. when fixed entries are set to zero), we can directly work in the framework of Lim et al. [39] that connects parameter symmetries to computation graph automorphisms. To work towards generalizing our result to $\kappa > 0$, we would have to modify the definitions and results of Lim et al. [39]; for instance, we would need to add edges associated to untrainable parameters in the computation graph, and redefine the concept of neural DAG automorphisms. We leave such exploration to future work.

**Theorem 3.** *If each mask matrix $M$ has unique nonzero rows, then $\mathbf{W}$-Asymmetric MLPs with $\kappa = 0$ have no nontrivial neural DAG automorphisms.*

*Proof.* Consider an $L$-layer $\mathbf{W}$-Asymmetric MLP with fixed entries set to zero. Denote its weights as $\mathbf{W}_L, \ldots, \mathbf{W}_1$ and the corresponding binary masks as $M_L, \ldots, M_1$. The forward pass of such a network on an input $x$ is then

$$[\mathbf{W}_L \odot M_L]\sigma(\cdots \sigma([\mathbf{W}_1 \odot M_1]x)\cdots), \tag{5}$$

for some elementwise nonlinearity $\sigma$. The dimension of $\mathbf{W}_i$ is $d_i \times d_{i-1}$. In the framework of Lim et al. [39], this is a feedforward neural network with a computation graph defined as follows.

The node set is $V_0 \times V_1 \times \ldots \times V_L$, where $V_i$ has $d_i$ nodes, and no nodes are shared between different $V_i$. If a node $v$ is in $V_i$, then we say that $\mathrm{layer}(v) = i$. $V_0$ contains the input nodes and $V_L$ contains the output nodes. The adjacency matrix can be written as

$$A = \begin{bmatrix} 0 & & & & \\ M_1 & 0 & & & \\ & M_2 & & & \\ & & \ddots & 0 & \\ & & & M_L & 0 \end{bmatrix}. \tag{6}$$

Every block besides the ones containing masks is zero. There are $L+1 \times L+1$ blocks, and the $(i,j)$ block is of size $d_i \times d_j$.

Recall that a neural DAG automorphism is a relabelling of nodes $\tau : V \to V$ such that $\tau$ is bijective, $(i,j) \in E$ if and only if $(\tau(i), \tau(j)) \in E$, and every input node and output node is a fixed point of $\tau$.

Now, let $\tau : V \to V$ be a neural DAG automorphism. Further, let $P$ be the corresponding permutation matrix. We will show that $\tau$ is the identity, i.e. that $P = I$. By Lemma 1, we know that $\tau$ preserves layer number of nodes, meaning $\mathrm{layer}(\tau(i)) = \mathrm{layer}(i)$. Thus, $P$ is a block diagonal permutation matrix:

$$P = \begin{bmatrix} P_0 & & & \\ & P_1 & & \\ & & \ddots & \\ & & & P_L \end{bmatrix}, \tag{7}$$

where $P_i$ is $d_i \times d_i$. Morever, $P_0 = I$ and $P_L = I$ because input nodes and output nodes are fixed points. Applying this to the adjacency matrix, we see that

$$\tau(A) = PAP^\top = \begin{bmatrix} 0 & & & \\ P_1 M_1 P_0^\top & 0 & & \\ & & \ddots & \\ & & P_L M_L P_{L-1}^\top & 0 \end{bmatrix}. \tag{8}$$

Since $\tau$ is a neural DAG automorphism, we have that $\tau(A) = A$. Equating blocks, this means that $P_1 M_1 P_0^\top = M_1$. As $P_0 = I$, we have $P_1 M_1 = M_1$. But $M_1$ has unique rows, so $P_1 = I$ as well.

For the inductive step, assume $P_i = I$ for some $i$. Then $P_{i+1}M_{i+1}P_i^\top = P_{i+1}M_{i+1} = M_{i+1}$, so since $M_{i+1}$ has unique rows, we have that $P_{i+1} = I$. As this holds for any $i$ by induction, this means that $P = I$, so $\tau$ is a trivial neural DAG automorphism and we are done. $\square$

**Lemma 1.** *Neural DAG automorphisms preserve layer number in* $\mathbf{W}$*-Asymmetric MLPs that have masks with nonzero rows.*

*Proof.* Let $\tau$ be a neural DAG automorphism. This means that $PAP^\top = A$, where $P$ is the permutation matrix associated to $\tau$. Then, using $PAP^\top = A$ for the first equality and the definition of $P$ in the second, we have that

$$A_{\tau(i),\tau(j)} = (PAP^\top)_{\tau(i),\tau(j)} = A_{i,j}. \tag{9}$$

We proceed by induction on layer number $l$. For any input node $i$ we know that $\tau(i) = i$, so of course $\mathrm{layer}(\tau(i)) = \mathrm{layer}(i)$.

Now, suppose that $\mathrm{layer}(\tau(i)) = \mathrm{layer}(i)$ for any $i$ in layer $l \geq 1$. If node $j$ is in layer $l + 1$, then there is some $i$ in layer $l$ such that $(i, j) \in E$ because $M_{l+1}$ has no nonzero rows. We have that $A_{\tau(i),\tau(j)} = A_{i,j}$, so $\tau(i)$ is connected to $\tau(j)$. As we know that $\tau(i)$ is in layer $l$, we have that $\tau(j)$ is in layer $l + 1$. $\square$

## B.2  Symmetry Breaking via Nonlinearities

**Proposition 3.** *Let the parameter space* $\Theta$ *be all pairs of square invertible matrices* $\theta = (\mathbf{W}_2, \mathbf{W}_1)$ *for* $\mathbf{W}_2, \mathbf{W}_1 \in GL(d)$*, and let* $f_\theta(x) = \mathbf{W}_2\sigma(\mathbf{W}_1 x)$*. If* $\sigma$ *has no linear equivariances, then* $f_{\theta_1} = f_{\theta_2}$ *if and only if* $\theta_1 = \theta_2$*. In other words, there are no nontrivial parameter space symmetries.*

*Proof.* If $\theta_1 = \theta_2$, then clearly $f_{\theta_1} = f_{\theta_2}$. For the other direction, suppose $f_{\theta_1} = f_{\theta_2}$, and denote $\theta_1 = (\mathbf{W}_2, \mathbf{W}_1)$ and $\theta_2 = (\widetilde{\mathbf{W}}_2, \widetilde{\mathbf{W}}_1)$. Then for any input $z \in \mathbb{R}^n$, we have

$$\mathbf{W}_2\sigma(\mathbf{W}_1 z) = \widetilde{\mathbf{W}}_2\sigma(\widetilde{\mathbf{W}}_1 z) \tag{10}$$

$$\widetilde{\mathbf{W}}_2^{-1}\mathbf{W}_2\sigma(\mathbf{W}_1 z) = \sigma(\widetilde{\mathbf{W}}_1 z). \tag{11}$$

Now, choose an arbitrary $x \in \mathbb{R}^n$. We let $z$ in the above equation (11) be $\mathbf{W}_1^{-1}x$, so we have

$$\widetilde{\mathbf{W}}_2^{-1}\mathbf{W}_2\sigma(x) = \sigma(\widetilde{\mathbf{W}}_1\mathbf{W}_1^{-1}x). \tag{12}$$

This holds for any $x$, so $\widetilde{\mathbf{W}}_2^{-1}\mathbf{W}_2 \circ \sigma = \sigma \circ \widetilde{\mathbf{W}}_1\mathbf{W}_1^{-1}$, i.e. we have found a linear equivariance of $\sigma$. Since $\sigma$ has no linear equivariances,

$$\widetilde{\mathbf{W}}_2^{-1}\mathbf{W}_2 = I = \widetilde{\mathbf{W}}_1\mathbf{W}_1^{-1}, \tag{13}$$

meaning that $\widetilde{\mathbf{W}}_2 = \mathbf{W}_2$ and $\widetilde{\mathbf{W}}_1 = \mathbf{W}_1$, i.e. $\theta_1 = \theta_2$, so we are done. $\square$

### B.2.1  FiGLU nonlinearity proofs (Proposition 2)

Now, we study the properties of our FiGLU nonlinearity $\sigma(x) = \eta(\mathbf{F}x) \odot x$, where $\eta$ is the sigmoid function $\eta(x) = \frac{1}{1+e^{-x}}$. For proving Proposition 2, we want to prove that with probability 1 over samples of $\mathbf{F}$, $\sigma$ has no permutation or diagonal equivariances.

We say that $\sigma$ has *no permutation equivariances* if whenever $P_2 \circ \sigma = \sigma \circ P_1$ for permutation matrices $P_1$ and $P_2$, then $P_1 = P_2 = I$. Likewise, we say that $\sigma$ has *no diagonal equivariances* if whenever $B \circ \sigma = \sigma \circ A$ for invertible diagonal matrices $A$ and $B$, then $A = B = I$.

We will show that these two properties hold for any $\mathbf{F}$ that has no permutation symmetries and no zero entries. We say that $\mathbf{F}$ has *no permutation symmetries* if $P_2\mathbf{F}P_1 = \mathbf{F}$ for permutation matrices $P_1$ and $P_2$ implies that $P_1 = P_2 = I$. Note that if $\mathbf{F}$ has distinct entries, then it has no permutation symmetries. Thus, $\mathbf{F}$ satisfies both of these conditions with probability 1, since the set of matrices with nondistinct entries or with at least one zero entry are of Lebesgue measure zero, so they have zero probability under the Gaussian distribution. We now proceed to show that $\sigma$ has no permutation or diagonal equivariances under these conditions on $\mathbf{F}$.

**Proposition 4.** *If* $\mathbf{F}$ *is a square matrix with no permutation symmetries, then* $\sigma(x) = \eta(\mathbf{F}x) \odot x$ *has no permutation equivariances.*

*Proof.* Suppose $\sigma \circ P_1 = P_2 \circ \sigma$ for permutation matrices $P_1, P_2$. We will show that $P_1 = P_2 = I$. For any input $x$, we have

$$\eta(\mathbf{F}P_1 x) \odot P_1 x = P_2 \left[ \eta(\mathbf{F}x) \odot x \right] \tag{14}$$

$$P_2^\top \left[ \eta(\mathbf{F}P_1 x) \odot P_1 x \right] = \eta(\mathbf{F}x) \odot x \tag{15}$$

$$\eta(P_2^\top \mathbf{F}P_1 x) \odot P_2^\top P_1 x = \eta(\mathbf{F}x) \odot x, \tag{16}$$

where we used permutation equivariance of $\eta$, which acts elementwise. Let $x = e_i$, the standard basis vector that is $1$ in the $i$th coordinate and $0$ elsewhere. If $i$ is not a fixed point of the permutation $P_2^\top P_1$, then let $j$ be the index that it is mapped to. Then equation (16) gives that

$$\eta(P_2^\top \mathbf{F}P_1 e_i) \odot P_2^\top P_1 e_i = \eta(\mathbf{F}e_i) \odot e_i \tag{17}$$

$$\eta(P_2^\top \mathbf{F}P_1 e_i) \odot e_j = \eta(\mathbf{F}e_i) \odot e_i. \tag{18}$$

In the $i$th coordinate of this equality of vectors, we see that $\eta(\mathbf{F}e_i) = 0$, which is impossible, since $\eta$ is the sigmoid function. Thus, $i$ cannot be a fixed point of $P_2^\top P_1$, so $P_2^\top P_1 = I$ is the identity permutation. Now, let $x$ be an arbitrary vector with no zero entries. Equation (16) gives that

$$\eta(P_2^\top \mathbf{F}P_1 x) \odot x = \eta(\mathbf{F}x) \odot x. \tag{19}$$

Since $x$ is nonzero, we can divide by $x_i$ in the $i$th coordinate of this vector equality for each $i$ to get that

$$\eta(P_2^\top \mathbf{F}P_1 x) = \eta(\mathbf{F}x). \tag{20}$$

As $\eta$ is bijective,

$$P_2^\top \mathbf{F}P_1 x = \mathbf{F}x. \tag{21}$$

Because this holds for all $x$ with no zero entries (and in particular for a basis of the input space), we know that

$$P_2^\top \mathbf{F}P_1 = \mathbf{F} \tag{22}$$

as matrices. But since $\mathbf{F}$ has no permutation symmetries, we have that $P_1 = P_2 = I$, so we are done. $\qquad\square$

**Proposition 5.** *If* $\mathbf{F}$ *is a square matrix with no zero entries, then* $\sigma(x) = \eta(\mathbf{F}x) \odot x$ *has no diagonal equivariances.*

*Proof.* Let $A = \mathrm{Diag}(\alpha)$ and $B = \mathrm{Diag}(\beta)$ be invertible diagonal matrices, and suppose that $\sigma \circ A = B \circ \sigma$. We will show that $A = B = I$. For any input $x$, we have

$$\eta(\mathbf{F}[\alpha \odot x]) \odot (\alpha \odot x) = \beta \odot \left[ \eta(\mathbf{F}x) \odot x \right]. \tag{23}$$

Let $x = ce_i$, where $e_i$ is the $i$th standard basis vector and $c \neq 0$ is any nonzero number. Then

$$\eta(\mathbf{F}c\alpha_i e_i) \odot c\alpha_i e_i = \beta \odot \left[ \eta(c\mathbf{F}e_i) \odot ce_i \right]. \tag{24}$$

At the $i$th coordinate of this equality, we have

$$\eta(\mathbf{F}c\alpha_i e_i)_i c\alpha_i = \beta_i \eta(c\mathbf{F}e_i)_i c \tag{25}$$

$$\frac{\alpha_i}{\beta_i} = \frac{\eta(c\mathbf{F}e_i)_i}{\eta(\alpha_i c\mathbf{F}e_i)_i} \tag{26}$$

Thus, the right hand side is constant in $c$. We must have that $\alpha_i > 0$, because if not, then increasing $c$ would increase either the numerator or denominator and decrease the other, hence contradicting the equality (here we use that $\mathbf{F}$ has no zero entries, so $c\mathbf{F}e_i$ is nonzero in every entry). Thus, letting $c \to \infty$, we see that $\frac{\alpha_i}{\beta_i} = 1$, so $\alpha_i = \beta_i$. Plugging this back into Equation (26), we have

$$1 = \frac{\eta(c\mathbf{F}e_i)_i}{\eta(\alpha_i c\mathbf{F}e_i)_i} \tag{27}$$

$$\eta(\alpha_i c\mathbf{F}e_i)_i = \eta(c\mathbf{F}e_i)_i \tag{28}$$

$$\alpha_i c(\mathbf{F}e_i)_i = c(\mathbf{F}e_i)_i \tag{29}$$

$$\alpha_i = 1, \tag{30}$$

where in the third line we used the fact that $\eta$ is invertible. We have shown that $\alpha_i = \beta_i = 1$ for each $i$, so $A = B = I$ and we are done. $\qquad\square$

We note that the proofs of these two results about FiGLU are reminiscent of some proof techniques from Godfrey et al. [20], such as those used in their analysis of GELU nonlinearities.

## B.3 Proofs for Universal Approximation

Here, we prove the universal approximation result for our **W**-Asymmetric MLPs.

**Theorem 4.** *Let $\eta$ be any nonpolynomial elementwise nonlinearity with $\eta(x) - \eta(-x) = x$ (e.g. ReLU, GELU, swish), let $\Omega \subseteq \mathbb{R}^D$ be a compact domain, and let $f_{\text{target}} : \Omega \to \mathbb{R}$ be a continuous target function. Fix $\varepsilon > 0$ and $\delta > 0$.*

*There exists a width $n'$ such that for all $n > n'$, with probability $1 - \delta$, for a randomly sampled 4-layer **W**-Asymmetric MLP $f$ with $\eta$ nonlinearity, hidden dimensions $24n \to n \to 24n$, and $n_{\text{fix}} \in o(n^{1/4})$ hardwired entries per neuron, there will exist $\theta \in \Theta$ such that the **W**-Asymmetric MLP $f : \mathbb{R}^n \to \mathbb{R}$ approximates $f_{\text{target}}$ to $\varepsilon$:*

$$\left\| f_\theta([x; 0]) - f_{\text{target}}(x) \right\| < \varepsilon \text{ for all } x \in \Omega. \tag{31}$$

*Importantly, we require that the input to $f_\theta$ be padded with $n - D$ zeroes, so $[x; 0] \in \mathbb{R}^n$.*

### B.3.1 Proof sketch

To approximate $f_{\text{target}}$ to $\varepsilon > 0$, we will leverage the universal approximation for standard MLPs with nonlinearity $\eta$ to first obtain a standard 2-layer MLP that approximates $f_{\text{target}}$ to within $\varepsilon$, meaning $\left\| f_{\text{target}}(x) - f_{\text{MLP}}(x) \right\| < \varepsilon$ for all $x \in \Omega$. Then we will exactly represent $f_{\text{MLP}}$ using an Asymmetric Network $f$.

This will be done by approximating each linear map $W$ of $f_{\text{MLP}}$ by two layers of an Asymmetric network: $W_2' \circ \eta \circ W_1' = W$ for Asymmetric linear maps $W_2'$ and $W_1'$. For the sake of exposition, we will show how to do this first when both $W_2'$ and $W_1'$ have no Asymmetric mask (i.e. fitting a linear map $W$ using a standard two-layer $\eta$-MLP), then when only $W_2'$ has an Asymmetric mask, and finally when both $W_2'$ and $W_1'$ have an Asymmetric mask.

### B.3.2 Fitting a Linear Map with a Two-Layer Standard MLP

Let $W \in \mathbb{R}^{n \times n}$ be the target linear map, and let $B \in \mathbb{R}^{2n \times n}$ and $A \in \mathbb{R}^{n \times 2n}$ be parameters of a two-layer MLP, defined by $f_{A,B}(x) = A\eta(Bx)$. We will choose $A$ and $B$ such that $f_{A,B}(x) = Wx$ for all $x \in \mathbb{R}^n$.

Denote the $i$th row of $W$ by $W_i$, so that

$$W = \begin{bmatrix} W_0 \\ \vdots \\ W_{n-1} \end{bmatrix}, \qquad Wx = \begin{bmatrix} W_0 \cdot x \\ \vdots \\ W_{n-1} \cdot x \end{bmatrix} \tag{32}$$

We set $A$ and $B$ as follows, where $I_n$ is the $n \times n$ identity matrix:

$$A = I_n \otimes \begin{bmatrix} 1 & -1 \end{bmatrix} = \begin{bmatrix} 1 & -1 & & & & \\ & & 1 & -1 & & \\ & & & & \ddots & \ddots \\ & & & & & 1 & -1 \end{bmatrix} \qquad B = \begin{bmatrix} W_0 \\ -W_0 \\ W_1 \\ -W_1 \\ \vdots \\ W_{n-1} \\ -W_{n-1} \end{bmatrix}. \tag{33}$$

Then we can see that $f_{A,B}$ exactly computes the linear transformation $Wx$.

$$A\eta(Bx) = \begin{bmatrix} \eta(W_0 \cdot x) - \eta(-W_0 \cdot x) \\ \vdots \\ \eta(W_{n-1} \cdot x) - \eta(-W_{n-1} \cdot x) \end{bmatrix} = \begin{bmatrix} W_0 \cdot x \\ \vdots \\ W_{n-1} \cdot x \end{bmatrix} = Wx. \tag{34}$$

### B.3.3 Fitting a Linear Map with One Asymmetric and One Standard Linear Map

Let $n_{\text{fix}} > 0$ and let each row of $\mathbf{N} \in \{0, 1\}^{n \times 6n}$ have $n_{\text{fix}}$ entries equal to 0, selected at random. Let $B \in \mathbb{R}^{6n \times n}$ and $A \in \mathbb{R}^{n \times 6n}$. Define $A'$ to be an Asymmetric linear map: $A' = A \odot \mathbf{N} + (1 - \mathbf{N}) \odot \mathbf{P}$, where $\mathbf{N}$ is a randomly sampled binary mask, and $\mathbf{P}$ a randomly sampled Gaussian matrix. We consider a two-layer network with one Asymmetric and one standard linear map: $f_{A,B}(x) = A'\eta(Bx)$. We want $f_{A,B}(x) = Wx$ for all x. For the remainder of this proof, we will assume that $\mathbf{N}$ never has three consecutive entries in a row set to zero; we will later show that this holds with high probability over the sampling of $\mathbf{N}$.

First, we define $B$ in a similar way to the purely linear setting, but with additional copies of entries to allow for error correction of the random noisy entries fixed in $A'$.

$$
B = \begin{bmatrix}
W_0 \\
W_0 \\
W_0 \\
-W_0 \\
-W_0 \\
-W_0 \\
\vdots \\
W_{n-1} \\
W_{n-1} \\
W_{n-1} \\
-W_{n-1} \\
-W_{n-1} \\
-W_{n-1}
\end{bmatrix}. \tag{35}
$$

Recall that each row $A'_i$ of $A'$ has $n_{\text{fix}}$ entries that are randomly hardwired to constants. Ideally, we would want $A'_i$ to pick out $\eta(W_i \cdot x) - \eta(-W_i \cdot x) = W_i \cdot x$, but because of the hardwired constants, $A'_i$ might randomly add $c * \eta(W_j \cdot x)$. However, since there are three copies of $\eta(W_j \cdot x)$ in $\eta(Bx)$, as long as not all three corresponding entries in $A'_i$ are fixed, one of the un-fixed copies can be changed such that the coefficients sum to 0. Since by our assumption $\mathbf{N}$ never has three consecutive entries all set to 0, the coefficients of $A$ can be picked such that $A'\eta(Bx) = Wx$. For example, a possible $A'$ matrix would be

$$
A' = \begin{bmatrix}
1 & 0 & 0 & -1 & 0 & 0 & | & \mathbf{1.1} & \mathbf{.37} & -1.47 & 0 & 0 & 0 & |\ldots & | & 0 & 0 & 0 & 0 & 0 & 0 \\
.89 & -\mathbf{.89} & 0 & 0 & 0 & 0 & | & \mathbf{.37} & .63 & 0 & -1 & 0 & 0 & |\ldots & | & 0 & 0 & 0 & 0 & 0 & 0 \\
0 & 0 & 0 & 0 & 0 & 0 & | & 0 & 0 & 0 & 0 & 0 & 0 & |\ldots & | & 1 & 0 & 0 & -1 & 0 & 0
\end{bmatrix}
$$

Thus we have shown that under the assumption that $\mathbf{N}$ never has three consecutive entries equal to 0, $A$ can be picked such that $A'\eta(Bx) = Wx$. We will now show that $\mathbb{P}(\mathbf{N}$ never has three consecutive entries equal to 0) can be made arbitrarily small by increasing the width $n$ while keeping $n_{\text{fix}}$ to be $o(n^{1/3})$.

The probability there are three consecutive entries in a given row of $\mathbf{N}$ that are zero is $O(\frac{n_{\text{fix}}^3}{n^2})$. By the union bound, the probability that any row has 3 consecutive hardwired entries is $O(\frac{n_{\text{fix}}^3}{n})$. For any $n_{\text{fix}} \in o(n^{1/3})$, this tends towards 0. Thus, with probability $\geq 1 - O(\frac{n_{\text{fix}}^3}{n})$, $A$ can be picked such that $A'\eta(Bx) = Wx$.

### B.3.4 Fitting a Linear Map with Two Asymmetric Linear Maps

Once again let $n_{\text{fix}} > 0$, and let each row of $\mathbf{M}$ have $n_{\text{fix}}$ entries equal to 0, selected at random. Let $B \in \mathbb{R}^{24n \times n}$ and $A \in \mathbb{R}^{n \times 24n}$. Further, define Asymmetric maps

$$
A' = A \odot \mathbf{N} + (1 - \mathbf{N}) \odot \mathbf{P}, \qquad B' = B \odot \mathbf{M} + (1 - \mathbf{M}) \odot \mathbf{Q}, \tag{36}
$$

where $\mathbf{N}, \mathbf{M}$ are randomly sampled masks, and $\mathbf{P}, \mathbf{Q}$ are normal matrices. Then we let $f_{A,B}(x) = A'\eta(B'(x))$, and we once again desire choices of parameters $A$ and $B$ such that $f_{A,B}(x) = Wx$.

**Constructing $B$**

Consider the randomly drawn mask $\mathbf{M} \in \{0,1\}^{24n \times n}$, and denote the $i$th row by $M_i$.

$$\mathbf{M} = \begin{bmatrix} M_0 \\ M_1 \\ \vdots \\ M_{24n-1} \end{bmatrix} \tag{37}$$

where $M_i \in \{0,1\}^n$. We partition $\mathbf{M}$'s rows into $n$ blocks of 24 rows. $\beta_1 = \{M_0 \ldots M_{23}\}, \ldots \beta_i = \{M_{24i} \ldots M_{24i+23}\}$. Now, consider $\beta_1$, the first 24 rows of $\mathbf{M}$.

**Definition B.1.** We say two rows $M_j$, $M_k$ are **intersecting** if there is some column index $\alpha$ such that $M_{j,\alpha} = M_{k,\alpha} = 0$. That is, two rows of are intersecting if they share a 0 at the same index.

Note that for any two given rows of $M$, the probability that they share a 0 in the same location is $\leq \frac{n_{\text{fix}}^2}{n}$.

We assume that $\beta_i$ contains no more than one pair of intersecting rows; later, we show this to hold with high probability. Then, every $\beta_i$ can be broken into two disjoint sets of 12 rows, $\beta_{i,0}$ and $\beta_{i,1}$, such that neither set of 12 contains a single pair of intersecting rows. Intuitively, this means that each row in $B$ corresponding to $\beta_{i,0}$ will have unique fixed indices.

Our goal will be for the rows in $\beta_i$ to mimic the row $W_i$. We will show how to do this for each $i$. Fix an arbitrary index $i \in \{0, \ldots, n-1\}$.

Without loss of generality, assume $\beta_{i,0}$ and $\beta_{i,1}$ are continuous, so $\beta_{i,0} = M_{24i:24i+11}$ and $\beta_{i,1} = M_{24i+11:24i+23}$. By our assumption, for $j, k \in \beta_{i,0}$ (i.e. $j, k \in \{0, \ldots, 11\}$), the mask rows $M_{24i+j}$ and $M_{24i+k}$ are never 0 in the same two column indices. Similarly, for $j, k \in \beta_{i,1}$ (i.e. $j, k \in \{12, \ldots, 23\}$), the mask rows $M_{24i+j}$ and $M_{24i+k}$ are never 0 in the same two column indices.

Next, we define $c_{i,j}$ as the difference between $B'$ and $B$ in the $(24i+j)$th row:

$$c_{i,j} = B'_{24i+j} - B_{24i+j}. \tag{38}$$

In particular, we have that

$$c_{i,j} = -B_{24i+j} \odot (1 - M_{24i+j}) + (1 - M_{24i+j}) \odot Q_{24i+j}. \tag{39}$$

**Lemma 2.** *For any indices $j \neq k$ such that $j, k \in \beta_{i,0}$ or $j, k \in \beta_{i,1}$, we have that*

$$c_{i,j} \odot M_{24i+k} = c_{i,j}. \tag{40}$$

*Proof.* By the definition of $c_{i,j}$, we know that $c_{i,j}$ is only nonzero at indices where $M_{24i+j}$ is equal to zero. Since $j, k$ are either both in $\beta_{i,0}$ or $\beta_{i,1}$, we know that $M_{24i+k}$ cannot also be zero at indices where $M_{24i+j}$ is zero. Thus, $M_{24i+k}$ is equal to 1 at every index where $c_{i,j}$ is nonzero, so $c_{i,j} \odot M_{24i+k} = c_{i,j}$ as desired. $\square$

Next, we construct $B$, by constructing this block of 24 rows. Let $[\mathbf{c}_{i,0}, \mathbf{c}_{i,1}, \mathbf{c}_{i,2}, \mathbf{c}_{i,3}]$ be continuous sums of length-3 segments of $c_{i,:}$:

$$\mathbf{c}_{i,0} = c_{i,0} + c_{i,1} + c_{i,2} \tag{41}$$
$$\mathbf{c}_{i,1} = c_{i,3} + c_{i,4} + c_{i,5} \tag{42}$$
$$\mathbf{c}_{i,2} = c_{i,6} + c_{i,7} + c_{i,8} \tag{43}$$
$$\mathbf{c}_{i,3} = c_{i,9} + c_{i,10} + c_{i,11} \tag{44}$$

We assign the first 12 rows of $B$ as follows.

$$(0 \leq j < 3) \to B_{24i+j} = W_i + \mathbf{c}_{i,0} - \mathbf{c}_{i,1} + \mathbf{c}_{i,2} - \mathbf{c}_{i,3} - c_{ij} \tag{45}$$
$$(3 \leq j < 6) \to B_{24i+j} = -W_i - \mathbf{c}_{i,0} + \mathbf{c}_{i,1} - \mathbf{c}_{i,2} + \mathbf{c}_{i,3} - c_{ij} \tag{46}$$
$$(6 \leq j < 9) \to B_{24i+j} = +\mathbf{c}_{i,0} - \mathbf{c}_{i,1} + \mathbf{c}_{i,2} - \mathbf{c}_{i,3} - c_{ij} \tag{47}$$
$$(9 \leq j < 12) \to B_{24i+j} = -\mathbf{c}_{i,0} + \mathbf{c}_{i,1} - \mathbf{c}_{i,2} + \mathbf{c}_{i,3} - c_{ij} \tag{48}$$

Defining $\mathbf{c} = \mathbf{c}_{i,0} - \mathbf{c}_{i,1} + \mathbf{c}_{i,2} - \mathbf{c}_{i,3}$, we have the nice property:

$$(0 \le j < 3) \to B'_{24i+j} = W_i + \mathbf{c} \tag{49}$$
$$(3 \le j < 6) \to B'_{24i+j} = -W_i - \mathbf{c} \tag{50}$$
$$(6 \le j < 9) \to B'_{24i+j} = +\mathbf{c} \tag{51}$$
$$(9 \le j < 12) \to B'_{24i+j} = -\mathbf{c} \tag{52}$$

By the construction above, $B'_{24i} = -B'_{24i+3}$, and $B'_{24i+6} = -B'_{24i+9}$. This means that

$$\eta(B'_{24i} \cdot x) - \eta(B'_{24i+3} \cdot x) = B'_{24i} \cdot x = (W_i + \mathbf{c}) \cdot x \tag{53}$$

and likewise that

$$\eta(B'_{24i+6} \cdot x) - \eta(B'_{24i+9} \cdot x) = \mathbf{c} \cdot x \tag{54}$$

So that a simple linear map gives our desired output:

$$\eta(B'_{24i} \cdot x) - \eta(B'_{24i+3} \cdot x) - [\eta(B'_{24i+6} \cdot x) - \eta(B'_{24i+9} \cdot x)] = (W_i + \mathbf{c} - \mathbf{c}) \cdot x \tag{55}$$
$$= W_i \cdot x. \tag{56}$$

In the next part, we will construct $A'$ to compute this linear map, which will follow the method of Appendix B.3.3 (because $A'$ has certain fixed entries).

What remains is to define the rows of $B$ corresponding to $\beta_{i,1}$ in an error correctible manner. This can be done easily by defining

$$\mathbf{d} = \sum_{j=12}^{23} c_{ij} \tag{57}$$

and then defining

$$(12 \le j < 24) \to B_{24i+j} = \mathbf{d} - c_{ij} \tag{58}$$

By similar reasoning to before, this means that

$$(12 \le j < 24) \to B'_{24i+j} = \mathbf{d}. \tag{59}$$

Recall that we constructed $B'$ under the assumption that no $\beta_i$ has at most one pair of intersecting rows. We now show that the $\beta_i$ each have at most one pair of intersecting rows with high probability. Within the 24 rows of any given $\beta_i$, the probability that more than one pair of rows are intersecting is $\le C\frac{n_{\text{fix}}^4}{n^2}$ for some constant $C$. So, by the union bound, the probability over $M$ that any of the $\beta_i$ have more than one pair of intersecting rows is $\le C\frac{n_{\text{fix}}^4}{n}$. Thus, we can construct $B$ in this manner with probability $\ge 1 - C\frac{n_{\text{fix}}^4}{n}$. For sufficiently large $n$ and $n_{\text{fix}} \in o(n^{1/4})$, this probability approaches 1.

**Construction of $A$**

With our above construction, each block of the 24 rows in $\beta_i$ of $\eta(B'x)$ is of the form

$$\begin{bmatrix} \eta((W_i + \mathbf{c}) \cdot x) \\ \eta((W_i + \mathbf{c}) \cdot x) \\ \eta((W_i + \mathbf{c}) \cdot x) \\ \eta((-W_i - \mathbf{c}) \cdot x) \\ \eta((-W_i - \mathbf{c}) \cdot x) \\ \eta((-W_i - \mathbf{c}) \cdot x) \\ \eta(\mathbf{c} \cdot x) \\ \eta(\mathbf{c} \cdot x) \\ \eta(\mathbf{c} \cdot x) \\ \eta(-\mathbf{c} \cdot x) \\ \eta(-\mathbf{c} \cdot x) \\ \eta(-\mathbf{c} \cdot x) \\ \eta(\mathbf{d} \cdot x) \\ \eta(\mathbf{d} \cdot x) \\ \eta(\mathbf{d} \cdot x) \\ \eta(\mathbf{d} \cdot x) \\ \eta(\mathbf{d} \cdot x) \\ \eta(\mathbf{d} \cdot x) \\ \eta(\mathbf{d} \cdot x) \\ \eta(\mathbf{d} \cdot x) \\ \eta(\mathbf{d} \cdot x) \\ \eta(\mathbf{d} \cdot x) \\ \eta(\mathbf{d} \cdot x) \\ \eta(\mathbf{d} \cdot x) \end{bmatrix} \tag{60}$$

Importantly, each row here is $n$ wide. Recall that $A' \in \mathbb{R}^{n \times 24n}$. Denote the $i$th row of $A'$ by $A'_i \in \mathbb{R}^{24n}$ with $A'_i \in \mathbb{R}^{24n}$. If $A'$ had 0 hardwired entries, then setting $A_i = \mathbb{1}_{24i} - \mathbb{1}_{24i+3} - (\mathbb{1}_{24i+6} - \mathbb{1}_{24i+9})$ would give $A_i\eta(B' \cdot x) = W_i \cdot x$, by the same argument as in Appendix B.3.2.

Unfortunately this is not the case, so we have to use the construction in Appendix B.3.3. Recall, that $A'$ has $n_{\text{fix}}$ fixed entries in each row. This means that $N_i$ has $n_{\text{fix}}$ entries equal to 0. Since every entry of $B'x$ has three copies, as long as $N_i$ does not have three elements set to 0 in a row, $A'_i$ can be made equivalent to $A_i = \mathbb{1}_{24i} - \mathbb{1}_{24i+3} - (\mathbb{1}_{24i+6} - \mathbb{1}_{24i+9})$. This is because, as in Appendix B.3.3, if at most 2 elements out of any 3 three copies are hardwired, then the third can be changed arbitrarily to offset the hardwiring.

Further, just as in Appendix B.3.3, the probability that a given row of $A'$ has three items hardwired in a row is $O(\frac{n_{\text{fix}}^3}{n^2})$. Thus, by the union bound, the probability that some row of $A'$ has three items hardwired in a row is $O(\frac{n_{\text{fix}}^3}{n})$. So, with large enough width $n$, $A$ can be chosen such that $A'\eta(B'x) = Wx$.

Similarly, any linear map in $\mathbb{R}^{\tilde{n} \times n}$ for $\tilde{n} < n$ can also be fit using this method.

**Conclusion**

We have shown that a $\mathbf{W}$-Asymmetric MLP with hidden dimension $24n$ can exactly fit an $n \times n$ linear map with high probability over the choice of Asymmetric masks $M$. It is known by [8] that for any continuous function $f_{\text{target}} : \Omega \subseteq \mathbb{R}^D \to \mathbb{R}$ and any $\varepsilon > 0$, there exists a width $k'$ such that 2-layer MLPs of width $k'$ can approximate $f$ to within $\varepsilon$.

Let $k$ be sufficiently big so that the probability that the masks do not satisfy the conditions of Appendix B.3.4 is less than $\delta$. Such a $k$ exists as long as $n_{\text{fix}} \in o(k^{\frac{1}{4}})$. Let $m \geq \max(k, k', D)$.

Importantly, if a 2-layer MLP of width $k'$ can approximate $f_{\text{target}}$ to within $\varepsilon$, a 2-layer MLP of width $m$ with $m \geq k'$ can also approximate $f$ to within $\varepsilon$. Let $f_{\text{MLP}}$ be a width $m$ MLP that approximates $f_{\text{target}}$ to within $\varepsilon$.

We now pad the input $x$ to $f_{\text{target}}$, with $m - D$ zeros. This allows us to define a new function $f^0_{\text{target}} : \mathbb{R}^m \to \mathbb{R}$ by $f^0_{\text{target}}([x; 0]) = f(x)$. Clearly $f^0_{\text{target}}$ can also be approximated by a width $m$ MLP.

Let $f^0_{\text{MLP}}$ denote the width $m$ MLP that approximates $f_0$ to within $\varepsilon$. Now, $f^0_{\text{MLP}}$ has dimensions $m \to m \to 1$, with corresponding linear maps $W_1 \in \mathbb{R}^{m \times m}$, $W_2 \in \mathbb{R}^{1 \times m}$. Each of these maps can be exactly fit using a 2-layer $\mathbf{W}$-Asymmetric MLP, since their corresponding matrices have at least as many columns as rows. Concatenating these two exact fits yields an asymmetric MLP whose output exactly matches $f^0_{\text{MLP}}$ and thus approximates $f_0$ to within $\varepsilon$.

Thus, setting $n' = m$, there exists a width $n'$ such that for all $n > n'$, with probability $1 - \delta$, for a randomly sampled 4-layer $\mathbf{W}$-Asymmetric MLP $f$ with $\eta$ nonlinearity, hidden dimensions $24n \to n \to 24n$, and $n_{\text{fix}} \in o(n^{1/4})$ hardwired entries per neuron, there will exist $\theta \in \Theta$ such that the $\mathbf{W}$-Asymmetric MLP $f : \mathbb{R}^n \to \mathbb{R}$ approximates $f_{\text{target}}$ to $\varepsilon$.

**On parameter symmetries of the 4-layer W-Asymmetric Network**

As an aside, this procedure of mapping 2-layer standard MLPs to 4-layer $\mathbf{W}$-Asymmetric MLPs implies that these 4-layer $\mathbf{W}$-Asymmetric MLPs have at least as many symmetries as 2 layer standard MLPs. To fix this, we may want to consider a nonlinearity $\eta$ such that $\eta(x) - \eta(-x) \neq x$.

## C    Limitations

Although our $\mathbf{W}$-Asymmetric and $\sigma$-Asymmetric networks are motivated by removing parameter space symmetries, their distinct empirical behavior may be caused by other factors besides just parameter space symmetries. For instance, the fixed entries $\mathbf{F}$ for the $\mathbf{W}$-Asymmetric approach are taken to be much larger than the standard initialization of linear maps, which could cause several changes to optimization and loss landscapes besides just parameter symmetry breaking.

Also, our theoretical results could be strengthened by future work in several ways. For instance, for the $\sigma$-Asymmetric approach, Proposition 1 only gives a guarantee of no parameter symmetries in the two-layer network case with square invertible weights. Future work could also give tighter analysis of the required width and depth for universal approximation using our $\mathbf{W}$-Asymmetric architecture.

## D    Broader Impacts

This work does not focus on any particular application area. Instead, we study fundamental phenomena and theory of deep learning in general. Our work has potential to improve known deficits of neural networks: by making neural network loss landscapes more similar to convex landscapes, we can improve our understanding of them, and by improving Bayesian neural networks we advance one paradigm for bettering uncertainty quantification in neural networks. However, unlike standard neural networks, which have millions of papers studying them, we have only scratched the surface of Asymmetric networks. Important properties such as generalization, robustness to distribution shifts, and adversarial robustness have not been extensively studied for Asymmetric networks, and the interaction of parameter symmetries with these properties is not clear. Future research should further explore these important properties.

## E    Experimental Ablations

Here, we present experiments with various ablations. These were in-part suggested by anonymous reviewers for the NeurIPS 2024 conference (thanks!).

### E.1    Matching Learnable Parameters

For the experiments in the main text, our $\mathbf{W}$-Asym networks often had less learnable parameters than the standard networks that they were compared against (since we take them to have the same architecture, but the $\mathbf{W}$-Asym approach fixes certain learnable parameters to constants). See Table 5 for the learnable parameter counts.

Table 5: Number of learnable parameters of Standard / $\sigma$-Asym and $\mathbf{W}$-Asym nets for our experiments.

| Experiment / Architecture | Standard / $\sigma$-Asym | W-Asym |
|---|---|---|
| 5.1 MLP | 935,434 | 834,570 |
| 5.1 ResNet 1x | 272,474 | 230,024 |
| 5.1 ResNet 8x | 17,289,866 | 16,273,946 |
| 5.1 GNN | 176,424 | 171,576 |
| 5.2 MLP-8 | 3,242,146 | 3,324,466 |
| 5.2 MLP-16 | 5,960,242 | 5,796,002 |
| 5.2 ResNet-20 1x | 1,356,098 | 1,143,858 |
| 5.2 ResNet-20 2x | 5,410,386 | 5,044,756 |
| 5.2 ResNet-110 1x | 8,620,418 | 7,371,378 |
| 5.2 ResNet-110 2x | 34,512,276 | 32,014,996 |
| 5.2 ResNet-20 2x | 5,410,386 | 5,044,756 |
| 5.3 ResNet | 78,042 | 60,634 |
| 5.4 MLI ResNet | 78,042 | 60,634 |

Table 6: Metanetwork performance, including results with smaller standard networks (Smaller ResNet) at same number of parameters at $\mathbf{W}$-Asym. There is no substantial difference — $\mathbf{W}$-Asym ResNets are still substantially easier to predict performance of.

| | ResNet | | Smaller ResNet | | W-Asym ResNet | |
|---|---|---|---|---|---|---|
| | $R^2$ | $\tau$ | $R^2$ | $\tau$ | $R^2$ | $\tau$ |
| MLP | $.330 \pm .04$ | $.389 \pm .03$ | $.348 \pm .07$ | $.400 \pm .02$ | $\mathbf{.594} \pm .12$ | $\mathbf{.864} \pm .01$ |
| DMC | $.950 \pm .01$ | $.787 \pm .02$ | $.943 \pm .01$ | $.779 \pm .01$ | $\mathbf{.967} \pm .01$ | $\mathbf{.911} \pm .01$ |
| DeepSets | $.855 \pm .01$ | $.617 \pm .03$ | $.849 \pm .01$ | $.627 \pm .01$ | $\mathbf{.936} \pm .00$ | $\mathbf{.858} \pm .00$ |
| StatNN | $.976 \pm .00$ | $.866 \pm .00$ | $.976 \pm .00$ | $.869 \pm .00$ | $\mathbf{.978} \pm .00$ | $\mathbf{.935} \pm .01$ |

In this section, we control for this, by decreasing the width of the standard networks so that they have the same number of parameters as the $\mathbf{W}$-Asymmetric networks they are compared against. Results are shown in Tables 6 and 7. We see that there is little to no change in metanetwork performance or Bayesian neural network performance for these smaller standard networks, so our results from the main text all still hold.

## E.2 Changing Number of Warmup Steps

Altintas et al. [3] showed that amount of learning rate warmup can affect the extent of linear mode connectivity between training runs. In Table 8, we see that varying the number of warmup epochs between 1 and 20 does not change the qualitative results much on test loss barrier for our linear mode connectivity experiments.

## E.3 Failure to Break Symmetries by Fixing Biases

Intuitively, one way to break permutation symmetries of an MLP is to order the hidden neurons in some way. One possible way to do this is to fix the biases (so that they are untrained). As shown in Table 9, a basic attempt at this fails, and has a much larger test barrier than our $\mathbf{W}$-Asym and $\sigma$-Asym networks.

# F Experimental Details

## F.1 Linear Mode Connectivity Experimental Details

### F.1.1 Image Classifier Interpolation

For the image classification experiments, we use two types of models.

Table 7: Bayesian NN test accuracy after 25 epochs. Decreasing standard ResNet20 parameters to match that of **W**-Asym ResNet20 does not substantially change performance.

| Base Network | Test Accuracy |
|---|---|
| **W**-Asym ResNet20 | $\mathbf{49.3} \pm 0.4$ |
| Standard ResNet20 | $46.8 \pm 0.9$ |
| Smaller Standard ResNet20 | $46.5 \pm 1.1$ |

Table 8: Test loss barrier when changing warmup steps. Results are very similar when lowering number of warmup epochs (**W**-Asym interpolates significantly better than Git-ReBasin). Adam optimizer with learning rate 1e-2 (ResNet20) and 1e-3 (GNN) is used.

| | ResNet20 (ReBasin) | **W**-Asym ResNet20 | GNN (ReBasin) | **W**-Asym GNN |
|---|---|---|---|---|
| 1 Epoch Warmup | $4.2 \pm .80$ | $\mathbf{.673} \pm .29$ | $.249 \pm .04$ | $.075 \pm .04$ |
| 20 Epoch Warmup | $2.0 \pm .21$ | $.934 \pm .72$ | $.292 \pm .04$ | $\mathbf{.074} \pm .02$ |

1. **ResNet** We train ResNet20s with LayerNorm of width $64$ and $8 \cdot 64$. We use a batch size of $128$ and a learning rate that warms up from .0001 to .01 over 20 epochs. In the width $8\times$ multiplier case we train for 50 epochs, and in the width $1\times$ multiplier case we train for 100. For $\sigma$-Asymmetric ResNets, we warm up to a learning rate of .001 instead of .01 due to training instability.

2. **MLP** We train MLPs with 4 layers, LayerNorm, and width 512. For MNIST we tuned the hyperparameters (epochs, learning rate, weight decay) of both the Asymmetric and Standard models to minimize loss barrier. We use a batch size of 64.

For MNIST we use no data augmentation, and for CIFAR-10 we use random cropping and horizontal flipping. For the Git-ReBasin tests, we use the weight matching algorithm from [1]. For MLPs on MNIST, we used the Asymmetry hyperparameters in Table 10. Table 11 gives the Asymmetric hyperparameters for ResNet20 on CIFAR-10, and Table 12 lists the same for ResNet20 with $8x$ larger width.

### F.1.2 Graph Neural Network Interpolation

For the GNN experiments, we use a GNN architecture similar to GIN [73] with mean aggregation. The base GNN has three message passing layers and a hidden dimension of 256, which gives 176,424 trainable parameters. The dataset is ogbn-arXiv [27], which is a citation network of computer science arXiv papers with 169,343 nodes and 1,166,243 edges. The task is transductive node classification, where the label of each paper node is the primary subject area of the paper.

As is common in transductive node classification on modestly sized graphs, we train each network with full-batch gradient on the whole graph. Thus, the randomness in training is purely from the initialization — there is no noise from minibatch selection in SGD. We use the Adam optimizer [31] with a peak learning rate of .001. The learning rate is linearly warmed up for 25 epochs to the peak, and then is held constant. Each network is trained for 500 epochs.

For the Git-ReBasin alignment, we implement the activation matching approach. For the $\sigma$-Asymmetric GNN, we take $\sigma$ to be FiGLU, in which we randomly initialize each fixed matrix $\mathbf{F}$ as a standard normal matrix with standard deviation $.01/\sqrt{d}$ where $d$ is the number of hidden channels; we found that having small standard deviation helped with training and interpolation. For the **W**-Asymmetric GNN, we fix 6 constants in each row of each linear map, and randomly initialize these constants from a normal distribution with standard deviation .5.

### F.2 Bayesian Neural Network Experimental Details

For training Bayesian neural networks, we use the variational inference approach of Tomczak et al. [64], which fits an approximate posterior that is Gaussian with a diagonal plus rank-4 covariance matrix structure. For the **W**-Asym ResNet tests, we train ResNet20s with the same Asymmetric hyperparameters as in Table 11, though with $\kappa = .5$. For the CIFAR-100 experiments, we use a

Table 9: Loss barrier when fixing biases to attempt to break symmetries for standard ResNet20 on CIFAR-10. Biases are between $[-k, k]$. Barriers are much larger than for $\mathbf{W}$-Asymmetric Nets $(.934 \pm .72)$ and $\sigma$-Asym Nets $(2.521 \pm .46)$.

| $k$ | Loss Barrier |
|---|---|
| 1 | $5.81 \pm 3.67$ |
| 3 | $3.76 \pm 0.38$ |
| 9 | $4.62 \pm 1.03$ |
| 27 | $10.6 \pm 4.29$ |

Table 10: $\mathbf{W}$-Asymmetric network hyperparameters for depth 4 MLPs. $n_{\text{fix}}$ refers to the number of weights we randomly fix per neuron. $\kappa$ refers to the standard deviation of the normal distribution that the fixed entries $\mathbf{F}$ are drawn from.

| Layer | $n_{\text{fix}}$ | $\kappa$ |
|---|---|---|
| Linear-1 | 64 | 1 |
| Linear-2 | 64 | 1 |
| Linear-3 | 64 | $\frac{1}{2}$ |
| Linear-4 | 256 | $\frac{1}{4}$ |

Table 11: $\mathbf{W}$-Asymmetric network hyperparameters for ResNet20s with width multiplier 1. $n_{\text{fix}}$ refers to the number of weights we randomly fix per output channel (for convolutional layers) or neuron (for linear layers). $\kappa$ refers to the standard deviation of the normal distribution that the fixed entries $\mathbf{F}$ are drawn from.

| Block | $n_{\text{fix}}$ | $\kappa$ |
|---|---|---|
| First Conv | 12 | 2 |
| Block 1 - Conv | 36 | 2 |
| Block 1 - Skip | 4 | 2 |
| Block 2 - Conv | 54 | 2 |
| Block 2 - Skip | 6 | 2 |
| Block 3 - Conv | 72 | 2 |
| Block 3 - Skip | 8 | 2 |
| Linear | 8 | 2 |

Table 12: $\mathbf{W}$-Asymmetric network hyperparameters for ResNet20s with width multiplier 8 on CIFAR-10. We use 3 times more fixed entries per output channel or neuron than for Table 11.

| Block | $n_{\text{fix}}$ | $\kappa$ |
|---|---|---|
| First Conv | 27 | 2 |
| Block 1 - Conv | 108 | 2 |
| Block 1 - Skip | 12 | 2 |
| Block 2 - Conv | 162 | 2 |
| Block 2 - Skip | 18 | 2 |
| Block 3 - Conv | 216 | 2 |
| Block 3 - Skip | 24 | 2 |
| Linear | 24 | 2 |

standard linear layer instead of hardwiring weights for the last fully-connected linear layer. On CIFAR-100 we also use a width multiplier of 2 for our ResNets. For the ResNet experiments, we use a learning rate of .001. We train with a batch size of 250 for 50 epochs.

For the MLP experiments, we use $\kappa = .5$, 8 hardwired entries per neuron, and a learning rate of .0005. A batch size of 250 is used for 50 epochs again.

We use standard data augmentation (horizontal flips and random crops) on CIFAR-10 and CIFAR-100, and no data augmentations for MNIST. All training is done with the Adam optimizer [31].

### F.3 Metanetwork Experimental Details

#### F.3.1 Dataset Details

We trained two datasets of image classifiers on CIFAR-10: one consisting of 10,000 small ResNet-like convolutional neural networks, and one consisting of 10,000 networks with a similar architecture, that use our graph-based approach to removing parameter symmetries. For fast training of many image classifiers, we use the FFCV package [37]. In particular, we use their CIFAR-10 sample script `https://github.com/libffcv/ffcv/tree/main/examples/cifar`, which includes data augmentation (random horizontal flips, random translations, and Cutout [10]), label smoothing [62], and a linear learning rate warmup and decay. In total, training all 20,000 classifiers takes just under 400 GPU hours (about 2 GPU-weeks) on NVIDIA RTX 2080 Ti GPUs.

See Table 13 for the hyperparameters and ranges that we varied across the networks in our datasets. In each dataset, the trained networks all have the same architecture.

Each ResNet has 78,042 trainable parameters, and each $\mathbf{W}$-Asym ResNet has 60,634 trainable parameters. Both have the same architecture, except the $\mathbf{W}$-Asym ResNet has certain filters that are fixed to constants to break the parameter symmetries. The ResNets each have 8 convolution layers, LayerNorm [5], and a final fully-connected linear classification layer after average pooling across spatial dimensions.

Table 13: Hyperparameters and distributions we sampled from for the datasets of image classifiers that we trained on CIFAR-10. $\mathrm{Unif}(a, b)$ is the uniform distribution over $[a, b]$, and $\mathrm{RandInt}(a, b)$ is the uniform distribution over integers in $[a, b]$ (inclusive of endpoints).

| Hyperparameter | Distribution |
|---|---|
| Learning rate | $.5 \cdot 10^{-\mathrm{Unif}(0,2)}$ |
| Weight decay | $10^{-\mathrm{Unif}(1,5)}$ |
| Label smoothing | $\mathrm{Unif}(0, .2)$ |
| Epochs | $\mathrm{RandInt}(10, 40)$ |

#### F.3.2 Metanetwork Details

Table 14: Learning rate and number of parameters for each type of metanetwork trained in Table 3.

| | ResNet | | $\mathbf{W}$-Asym ResNet | |
|---|---|---|---|---|
| | LR | # Params | LR | # Params |
| MLP | $10^{-4}$ | 4,994,945 | $10^{-4}$ | 3,880,833 |
| DMC [12] | $10^{-3}$ | 105,357 | $5 \cdot 10^{-3}$ | 105,357 |
| DeepSets [77] | $10^{-2}$ | 8,897 | $5 \cdot 10^{-3}$ | 8,897 |
| StatNN [65] | $10^{-3}$ | 119,297 | $10^{-2}$ | 119,297 |

We trained several types of metanetworks for our experiments. All of these metanetworks are trained for 50 epochs using the AdamW optimizer [40]. For each metanetwork, on each dataset, we choose the learning rate in $\{10^{-5}, 10^{-4}, 5 \cdot 10^{-4}, 10^{-3}, 5 \cdot 10^{-3}, 10^{-2}\}$ that gives the best validation $R^2$ performance on one training run. Then we run train each type of metanetwork 5 times on each dataset, and report the mean and standard deviation for each metric in Table 3.

### F.4 Monotonic Linear Interpolation Experimental Details

For the monotonic linear interpolation experiments, we used the same setup as in the training of the datasets of CIFAR-10 image classifiers in Section 5.3. For each architecture, we sample 300 sets of hyperparameters from the distributions in Table 13, and train one network for each set of these sampled hyperparameters. When evaluating training loss, we include the labeling smoothing term.

For the $\sigma$-Asymmetric networks, we initialize the FiGLU $\mathbf{F}$ with a standard deviation of $1/\sqrt{d}$, where $d$ is the number of channels in the layer. Note that this is considerably larger than the standard

deviation of $.01/\sqrt{d}$ used in the GNN experiments of Section 5.1; we found this setting to train better (note that this initialization is in line with standard initializations of trainable parameters). Further, for the $\sigma$-Asymmetric networks, 24 out of the 300 networks diverged during training (giving `NaNs`), so we exclude them from the computation of statistics in Table 4. From manual inspection, this divergence seems to happen when the learning rate is high (greater than .1). In contrast, none of the standard or **W**-Asymmetric networks diverged.

### F.5 Miscellaneous Experimental Details

The datasets we use are MNIST [38], CIFAR-10 [33], CIFAR-100 [33], and ogbn-arXiv [27], which are all widely used in machine learning research. The first three appear to not have licenses and are open to use, while the last dataset is from the Open Graph Benchmark, which has an MIT License in the Github repository.

We use software packages including PyTorch [52] (for all neural network experiments), FFCV [37] (for building our dataset in Section 5.3), and PyTorch Geometric [15] (for GNN experiments).

We ran our experiments on several types of NVIDIA GPUs and compute systems, including 2080 Ti, 3090 Ti, 4090 Ti, and V100 GPUs. Every training run was conducted on at most one GPU.

