# OpenReview forum: "The Empirical Impact of Neural Parameter Symmetries, or Lack Thereof"
_NeurIPS.cc/2024/Conference — NeurIPS 2024 poster_

### Official Review · Reviewer_tX9W · 2024-07-11

**Soundness:** 3
**Presentation:** 4
**Contribution:** 4
**Rating:** 7
**Confidence:** 4

**Summary:**

This paper develops two types of modifications to neural networks to remove permutation symmetries: fixing certain weights, or using a non-elementwise activation function. The resulting "asymmetric" neural networks are found to improve on certain metrics that are observed in networks with permutation symmetries, and thought to be caused by such symmetries: lack of linear mode connectivity between randomly initialized/trained networks, reduced Bayesian network performance, and reduced metanetwork performance. The asymmetric networks also are more likely to have monotonic linear interpolation between their initial and trained weights.

**Strengths:**

This paper is effectively an ablation study on permutation symmetries, which provides a much-needed tool and counterfactual perspective for investigations of permutation symmetries. The methods appear both novel and logical. A thorough survey of prior literature is given, which is extremely welcome. The paper is well organized, with well-motivated methodology and easy-to-follow experiments and hypotheses are easy to follow. The proofs are intuitive and comprehensive.

**Weaknesses:**

The main issue is that the paper relies heavily on experimental rather than theoretical results, but the strongest results are for a single method ($W$-asymmetry) with settings much more different than "standard" training than seems at first glance - specifically, lower learning rates, longer warmup, and larger fixed weights that all differ from standard training by order(s) of magnitude. These settings, plus a lack of available code, seriously impact the relevance/reproducibility of the methods/results.

There should be a comparison with standard models initialized to the same weights as the $W$-asymmetric networks. This would make certain metrics like $L^2$ weight distance comparable (currently the $W$-asymmetric networks are more likely to have extremely large magnitude weights which could increase the $L^2$ distance between independently initialized/trained networks). This would also eliminate some other variables such as reduced performance due to suboptimal weight initialization.

Low learning rates and long warmup are likely to improve similarity and reduce barriers, particularly between identically initialized neural networks. An analysis of how learning rate/warmup affects the experimental results (particularly barriers after training) is needed to determine if the observed results are mainly due to asymmetry.

Some additional details are needed to properly replicate the asymmetry methods, specifically in tables 5-7.
- how are $n_{\text{fix}}$ and $\kappa$ determined for each architecture?
- what do the names of the blocks refer to? There are not enough for a 1-to-1 mapping onto ResNet20 layers or permutations, so more explanation is needed.

**Questions:**

Are the results in table 1 for networks with different random initializations, or the same identical initialization? If the former, are the masks and fixed weights different or the same between the two networks? I find it highly unlikely that networks with different fixed weights or different initializations would have low barrier after training without post-processing.

The fixed weights in $W$-asymmetry are distributed very differently to the non-fixed weights. If the fixed linear transform $F$ in $\sigma$-asymmetry is similarly shifted in distribution, would that make its results closer to $W$-asymmetry?

Why not fix biases to get asymmetry instead?

**Limitations:**

The reproducibility issues discussed previously, combined with the experimentally-focused framing, detract from an otherwise very strong and impactful paper.

The results for $\sigma$-asymmetry are considerably weaker than $W$-asymmetry (and rely on even smaller learning rates). Another work has found that networks mainly use linear transformations [1]. Since $\sigma$-asymmetry is effectively a non-elementwise sigmoid activation function, I wonder if the network's activations mainly stay in the approximately linear region of the sigmoid, allowing many permutations to be approximately equivalent. Some other comments or experiments on why $\sigma$-asymmetry is less effective in practice would be helpful, especially since theoretically speaking, both methods are clearly asymmetric.

[1] Mehmeti-Göpel & Disselhoff. Nonlinear Advantage: Trained Networks Might Not Be As Complex as You Think. ICML 2023. https://proceedings.mlr.press/v202/ali-mehmeti-gopel23a.html

---

> ### Author Rebuttal · Authors · 2024-08-07
>
> We thank the reviewer for their appreciation of the novelty of our methods, thoroughness of our discussion of previous work, organization of the paper and methodology, and comprehensiveness of our proofs. Also, we thank the reviewer for putting effort into understanding the empirical setup of our work, and making suggestions to improve the $\sigma$-Asymmetric networks.
>
> **On Reproducibility:** As mentioned in the paper checklist, we plan to open source the data and code to the public, and we are sharing the code with the reviewers now. We have sent the AC our data and code to reproduce the experiments, which they will share with you.
>
> We have also run useful ablations that the reviewer suggested, and found that the results do not significantly change! In fact, some of the reviewer’s suggestions actually improved the results (e.g. shorter warmup period improved W-Asymmetry interpolation while making Git-ReBasin interpolation worse).
>
> > “The main issue is that the paper relies heavily on experimental rather than theoretical results, but the strongest results are for a single method (W-asymmetry) with settings much more different than "standard" training … lower learning rates, longer warmup, and larger fixed weights that all differ from standard training by order(s) of magnitude.”
>
> We have rerun our ResNet20 experiment on CIFAR10, and find that using a more standard warmup schedule (1 epoch) yields similar results for linear interpolation. In fact, using a 1 epoch warmup has a lower loss barrier (.691) than the 20 epoch warmup (.931). Performance is also similar on GNNs, see our 1-page results PDF for more.
>
> Our learning rates used for our ResNet training are pretty standard for the Adam optimizer (1e-2 peak). When using 1e-3 learning rate for W-Asymmetric nets, we actually get a lower barrier of $.285 \pm .065$, but test loss that is significantly worse. Perhaps the reviewer is referring to the higher learning rates that people use with SGD (such as 1e-1 in Git-ReBasin), but these learning rates are normal for Adam (e.g. Git-ReBasin uses 1e-3 for Adam on their MLP, and the folklore [3e-4 of Karpathy](https://karpathy.github.io/2019/04/25/recipe/) is common).
>
> Indeed the magnitude of the fixed weights we use are very large but that is by design. We note that we require high magnitude fixed weights in order to effectively break both approximate and exact parameter symmetries. Nevertheless, we agree that the large fixed weights are not ideal for W-Asymmetry. Our work is fairly novel, in that others have not really worked on making Asymmetric networks like this. We envision that future works will develop new types of Asymmetric networks that get around some of these issues.
>
> > “There should be a comparison with standard models initialized to the same weights as the W-asymmetric networks…
>
> Standard networks initialized to the same weights have similar performance ($.79 \pm .2$ loss barrier with 1 epoch warmup, versus $.67 \pm .3$ for W-Asym, for ResNet20 on CIFAR-10). We also find that these networks are comparably asymmetric though have slightly worse linear interpolation. We find this not too surprising since the symmetry-breaking weights have such high magnitudes and thus are similar before and after training.
>
> > “how are $n_{\mathrm{fix}}$  and $\kappa$ determined for each architecture? … what do the names of the blocks refer to? There are not enough for a 1-to-1 mapping onto ResNet20 layers or permutations, so more explanation is needed.”
>
> Our theory suggests that $n_{\mathrm{fix}}$ should be at least $\log_2(dim)$ so that the conditions in Theorem 3 hold. The exact values of $n_{\mathrm{fix}}$ and $\kappa$ are chosen manually (these are hyperparameters).
>
> The ResNet20 code we use is in `lmc/models/models_resnet.py` in our attached code. It consists of an initial convolution followed by 3 blocks. Each block contains 6 convolutions (and 36 convolutions for ResNet110). The parameters we give in Tables 6 and 7 apply to each convolution of the described block.
>
> > “Are the results in table 1 for networks with different random initializations, or the same identical initialization? … are the masks and fixed weights different or the same between the two networks? ...”
>
> The learnable parameters are from different random initializations, but the masks and fixed weights are the same between the two networks. See also the general comment for more on this.
>
> > “... If the fixed linear transform F in $\sigma$-asymmetry is similarly shifted in distribution, would that make its results closer to W-asymmetry?”
>
> Good question. We have tried this as well (as noted in the paper, we tune the standard deviation that $\mathbf{F}$ is drawn from for $\sigma$-Asymmetry as well), but this was not sufficient.
>
> > “Why not fix biases to get asymmetry instead?”
> We tried this and it didn’t work. See the results in our results PDF. Intuitively, there is only one dimension for the biases to vary, so for instance if you sort the biases, then two neighboring biases may be very close to each other. This causes two neurons to be approximately automorphic, so there are still approximate permutation symmetries.
>
> > “for $\sigma$-asymmetry … I wonder if the network's activations mainly stay in the approximately linear region of the sigmoid … Some other comments or experiments on why $\sigma$-asymmetry is less effective in practice would be helpful …”
>
> We thank the reviewer for this suggestion. This is a plausible cause, but we investigated it, and it does not seem like this is the main issue. For instance, to make the nonlinearity more nonlinear, we switched to cosine nonlinearity instead of sigmoid, but this did not work well either. Also, we have already tried changing the magnitude of the fixed weights (via tuning $\kappa$), to move the activations into different parts of the sigmoid. See general comment for more.

---

> > ### Comment · Reviewer_tX9W · 2024-08-08
> >
> > Thank you for the detailed response and additional ablations, as well as releasing the source code.
> >
> > Apologies for the confusion regarding the learning rate, I did not notice you were using Adam. The ablations are very promising, and it is good to know the method does not depend on the warmup schedule. Although the performance between fixed and non-fixed W-asymmetric weights is not as different as with regular networks, it is still important to know that fixing the weights significantly reduces barriers, which proves the method's utility is not just in the differing weight initialization. I am willing to significantly raise my score after reviewing the source code.
> >
> > Thank you for entertaining the many possible explanations for differences between $W$- and $\sigma$-asymmetry. I believe such negative results are important both for building asymmetric experiments, and for understanding how networks train in practice.

---

> > > ### Author Response · Authors · 2024-08-12
> > >
> > > Hello Reviewer tX9W, we were wondering whether you have received the code from the AC, given that you wanted to take a look at it (we sent it as soon as we submitted the rebuttal). Let us know if there are any questions!

---

> > > > ### Comment · Reviewer_tX9W · 2024-08-13
> > > >
> > > > Hi, I've received the code and will update my score tomorrow. No issues so far.

---

> > > > > ### Comment · Reviewer_tX9W · 2024-08-13
> > > > >
> > > > > Hi, could the authors clarify whether the LMC results (e.g. table 1) use layer or batch normalization? I noticed the batch normalization is commented out in the code in favour of layer normalization.

---

> > > > > > ### Author Response · Authors · 2024-08-13
> > > > > >
> > > > > > Hello, the LMC results use layer norm, as is done in e.g. Git Rebasin. We will also update you with batch norm results for comparison.

---

> > > > > > > ### Comment · Reviewer_tX9W · 2024-08-14
> > > > > > >
> > > > > > > I have reviewed the code and raised my score accordingly. All the best and good luck!

---

### Official Review · Reviewer_ceUs · 2024-07-13

**Soundness:** 1
**Presentation:** 2
**Contribution:** 2
**Rating:** 5
**Confidence:** 4

**Summary:**

The paper suggested asymmetric neural networks in terms of an asymmetric nonlinearity and weight. It demonstrated some tasks including linear mode connectivity without permutation alignment, Bayesian neural networks, training metanetworks, and monotonic linear interpolation to show the role of permutation symmetry inherited in the standard neural networks.

**Strengths:**

1. Various evaluations to show the role of permutation symmetry.
1. Novel approaches to build neural networks that have no permutation symmetry.

**Weaknesses:**

1. The number of learnable parameters of the standard NN and the asymmetric NN is reported only for the metanetwork task. It should also be reported for the LMC, BMA, and MLI tasks.
1. Comprehensible reason is required for the proposition “the posterior will have less modes” in the hypothesis of the BMA task in order to understand the purpose of the BMA task. No symmetry does not imply less modes. Thus, the improved performance of BMA may be induced from another reason other than asymmetry.

**Questions:**

What do you mean by “loss” in BNN. Is it an ELBO? Why don’t you report the negative log-likelihood (NLL)?

**Limitations:**

1. Fixing some weights or adding anisotropic activation function implies constraining not only symmetry but also distorted parameter manifold (like equivariant NNs) that also leads to increasing correlation between solutions. The asymmetric NN is limited to directly correspond to the standard networks with a fixed permutation.
1. The BMA and MLI tasks are based on the strong assumption, where its evidence is absent, that no symmetry induces the reduced number of modes.
1. Although the paper partially gives insights on the loss surface, it does not suggest a new practical method utilizing the insights.

---

> ### Author Rebuttal · Authors · 2024-08-07
>
> We thank the reviewer for appreciating the novelty and evaluations behind our work, and for their comments, which we now address one at at time:
>
> > “The number of learnable parameters of the standard NN and the asymmetric NN is reported only for the metanetwork task. It should also be reported for the LMC, BMA, and MLI tasks.”
>
> Good point, we have now added the number of learnable parameters for the other experiments to our paper. They are in this table (note that sigma-Asym networks always have the same number of learnable parameters as the corresponding standard network).
>
> | Experiment / Architecture  | Standard / sigma-Asym | W-Asym|
> | --- | --- | --- |
> | Sec 5.1 MLP |  935434 | 834570  |
> | Sec 5.1 Resnet 1x  | 272,474 | 230,024 |
> | Sec 5.1 Resnet 8x  | 17,289,866 | 16,273,946 |
> | Sec 5.1 GNN | 176,424 | 171,576 |
> | Sec 5.2 MLP-8 | 3,242,146 | 3,324,466 |
> | Sec 5.2 MLP-16 | 5,796,002 | 5,960,242 |
> | Sec 5.2 Resnet-20 1x | 1,143,858 | 1,356,098 |
> | Sec 5.2 Resnet-20 2x | 5,044,756 | 5,410,386 |
> | Sec 5.2 Resnet-110 1x | 7,371,378 | 8,620,418 |
> | Sec 5.2 Resnet-110 2x | 32,014,996 | 34,512,276 |
> | Sec 5.2 Resnet-20 2x | 5,044,756 | 5,410,386 |
> | Sec 5.4 MLI ResNet | 78,042 | 60,634 |
>
>
> We also now report the number of learned parameters for metanetworks in Section 5.3 (not the input data) as follows:
>
> | Metanet | Num Params |
> | --- | --- |
> | MLP | 4,994,945 (ResNet) / 3,836,673 (Smaller ResNet) / 3,880,833 (W-Asym ResNet) |
> | DMC | 105,357 |
> | DeepSets | 8,897 |
> | StatNN | 119,297 |
>
> > “Comprehensible reason is required for the proposition “the posterior will have less modes” in the hypothesis of the BMA task in order to understand the purpose of the BMA task. No symmetry does not imply less modes… “
>
> > “The BMA and MLI tasks are based on the strong assumption, where its evidence is absent, that no symmetry induces the reduced number of modes.”
>
> We gave several citations in that section (5.2) on previous works noting that symmetries induce problematic modes in Bayesian NNs (e.g. [2, 33, 71, 49, 70, 35]).
>
> The argument is simple: if $\tau$ is a parameter symmetry, then $L(\theta) = L(\tau(\theta))$ for loss functions $L$ such as NLL and parameters $\theta$ (because the neural network function is left unchanged by $\tau$, and the loss only depends on the neural network function). Thus, for any mode $\theta^* \in \mathrm{argmin}_{\theta} L(\theta)$, we have that $\tau(\theta^*)$ is also a mode, because it has the same loss value. Removing parameter symmetries $\tau$ also removes these additional modes. We will spell out this argument in the revision.
>
> Also, the empirical results do not rely on these “assumptions”. But rather, previous work shows that we expect these assumptions to hold, so we empirically investigate these hypotheses via Asymmetric Networks.
>
> > “What do you mean by “loss” in BNN. Is it an ELBO? Why don’t you report the negative log-likelihood (NLL)?”
>
> We do mean NLL loss. We will make this clearer in the revision.
>
> > “Fixing some weights or adding anisotropic activation function implies constraining not only symmetry but also distorted parameter manifold (like equivariant NNs) that also leads to increasing correlation between solutions. The asymmetric NN is limited to directly correspond to the standard networks with a fixed permutation.”
>
> Regarding the comparison to equivariant NNs, coefficients for a steerable basis in an equivariant network is probably more distorting of the parameter manifold. Yes, we do distort the parameter manifold, but we do so in a way that maintains the vector space structure and calculus, so we can optimize Asymmetric Networks with standard gradient-based methods like Adam. In contrast, some previous works restrict parameters to nonlinear manifolds, and require different optimization algorithms such as projected gradient descent or Riemannian optimization methods. See the general comment for more information, as well as our related work section.
>
> >  “The asymmetric NN is limited to directly correspond to the standard networks with a fixed permutation.”
> We are not sure what you mean here, but this is an important positive property of asymmetric networks. Since they are like standard networks with a fixed permutation, we do not need to account for parameter symmetries when doing interpolation, metanetwork processing, or Bayesian NN training for asymmetric networks.
>
> > “Although the paper partially gives insights on the loss surface, it does not suggest a new practical method utilizing the insights.”
>
> This is not quite the purpose of our paper; we moreso seek to understand various phenomena in deep learning. Some phenomena like Monotonic Linear Interpolation (MLI) have no real applications, and some others like Linear Mode Connectivity (LMC) have related downstream applications in things like model merging and federated model averaging.
>
> Moreover, the BNN results are in some sense a practical method for improving training of BNNs.
>
> That being said, we do believe there are some other potential practical methods inspired by our work, which could be explored in future work. For instance, model merging is extremely powerful for open-weight large language models (see top methods on [open LLM leaderboard](https://huggingface.co/spaces/open-llm-leaderboard-old/open_llm_leaderboard)), and symmetries have not been explored as much there. Our Asymmetry methods cause little overhead (only modifying weights and/or nonlinearities a little), so they could be used for very large models.

---

> > ### Comment · Reviewer_ceUs · 2024-08-12
> > **Rebuttal**
> >
> > Thank you very much for the detailed clarification. While I understood some of them, I still have concerns regarding the proposition.
> >
> > As you and the cited papers mentioned, when we fix an architecture (both in width and depth) and then fix some weights, the total number of modes will obviously be reduced. However, in a fair comparison, where the standard NN and the asymmetric NN have the same number of learnable parameters, the architectures, and thus their parameter spaces, will be different. Since we have no idea about the total number of modes for each architecture, it cannot be definitively stated that the number of modes will be reduced. In fact, the modes could potentially increase due to the added nodes. Furthermore, **the number of modes can be uncountably many** when ReLU nonlinearity (e.g., ResNet) is used due to its scaling symmetry. For these reasons, I believe a comprehensive (theoretical) justification is necessary.
> >
> > Although you claimed that the experiment does not rely on this assumption, I believe it does. If the number of modes is not reduced, the improved BNN performance in the asymmetric NN could simply be a result of the increased number of learnable parameters, as you mentioned in the rebuttal table. If you intend to demonstrate the hypothesis regarding the number of modes, I would recommend using the exact same architecture for both the standard NN and the asymmetric NN, even if it causes fewer parameters.

---

> > > ### Author Response · Authors · 2024-08-12
> > >
> > > Thanks for the clarification!
> > >
> > > We would like to clarify that we already have empirical results in **both of the experimental regimes that you discuss here**.
> > >
> > > 1. In all of the experiments of our original submission, when comparing a standard and Asymmetric network, both networks have the same exact base architecture (width, depth, modules), besides the fact that the Asymmetric network has either some weights fixed (W-Asymmetric), or uses FiGLU nonlinearities (with an additional F matrix). We believe this is what you desire when you say "I would recommend using the exact same architecture for both the standard NN and the asymmetric NN, even if it causes fewer parameters." **All experiments in our original submission already followed this recommendation!**
> > > 2. In new experiments included in the rebuttal, as suggested by Reviewer ajZW, we matched the number of parameters of the standard and Asymmetric networks for Bayesian NNs and Metanetworks, by making the standard networks have less width.
> > >
> > > Given that our empirical results are essentially the same in both regimes, we believe the empirical evidence is strong.
> > >
> > > > "the number of modes can be uncountably many when ReLU nonlinearity (e.g., ResNet) is used due to its scaling symmetry"
> > >
> > > In the case of continuous symmetries, we can use the dimension of the symmetry group (as a lie group) as an analogous measure of "number of modes". Then, since $\sigma$-Asymmetry removes scaling symmetries provably, and $\mathbf{W}$-Asymmetry removes any obvious scaling symmetries, we can again argue that there are less symmetries. We will be more careful with our wording around "less modes" in the revision.
> > >
> > >
> > > > "I believe a comprehensive (theoretical) justification is necessary."
> > >
> > > We agree with the reviewer that this would be nice. We only have a full understanding of the theory in the case considered in Proposition 3 (two-layer MLPs with square invertible weights), where we can say that, as long as only one neural network function is a minima of the loss, then there is exactly one mode of the loss landscape. Further theoretical results have been more difficult to derive, but we believe that future work could build on this.

---

> > > > ### Comment · Reviewer_ceUs · 2024-08-12
> > > >
> > > > You said W-Asym has more learnable parameters in the BNN task (Sec 5.2) as in the table below (brought from your first rebuttal) but W-Asym should have less parameters to have the same architecture with the standard NN. I don't understand why your experiments already follow the exact same architectures. Furthermore, I think that the standard and sigma-Asym are not the same architecture even when their learnable parameters are the same because of the additional fixed matrix inside of the non-linearity.
> > > >
> > > > | Experiment / Architecture | Standard / sigma-Asym | W-Asym     |
> > > > |---------------------------|-----------------------|------------|
> > > > | Sec 5.1 MLP               | 935434                | 834570     |
> > > > | Sec 5.1 Resnet 1x         | 272,474               | 230,024    |
> > > > | Sec 5.1 Resnet 8x         | 17,289,866            | 16,273,946 |
> > > > | Sec 5.1 GNN               | 176,424               | 171,576    |
> > > > | Sec 5.2 MLP-8             | 3,242,146             | 3,324,466  |
> > > > | Sec 5.2 MLP-16            | 5,796,002             | 5,960,242  |
> > > > | Sec 5.2 Resnet-20 1x      | 1,143,858             | 1,356,098  |
> > > > | Sec 5.2 Resnet-20 2x      | 5,044,756             | 5,410,386  |
> > > > | Sec 5.2 Resnet-110 1x     | 7,371,378             | 8,620,418  |
> > > > | Sec 5.2 Resnet-110 2x     | 32,014,996            | 34,512,276 |
> > > > | Sec 5.2 Resnet-20 2x      | 5,044,756             | 5,410,386  |
> > > > | Sec 5.4 MLI ResNet        | 78,042                | 60,634     |

---

> > > > ### Author Response · Authors · 2024-08-12
> > > >
> > > > Hello, looking at the comments again, we realize why there may be some confusion. In our original rebuttal, the Bayesian experiments of the number of learnable parameters table has a typo, and the two columns are switched.
> > > > In particular, in every row, the W-asym experiment should have fewer parameters (because as mentioned above, it has the same width, depth, and modules as the standard network). Here's the updated table. Sorry for any confusion this may have caused.
> > > >
> > > > |  Experiment / Architecture | Standard / sigma-Asym | W-Asym|
> > > > | --- | --- | --- |
> > > > | Sec 5.1 MLP |  935,434 | 834,570  |
> > > > | Sec 5.1 Resnet 1x  | 272,474 | 230,024 |
> > > > | Sec 5.1 Resnet 8x  | 17,289,866 | 16,273,946 |
> > > > | Sec 5.1 GNN | 176,424 | 171,576 |
> > > > | Sec 5.2 MLP-8 |3,324,466 | 3,242,146 |
> > > > | Sec 5.2 MLP-16 | 5,960,242 | 5,796,002 |
> > > > | Sec 5.2 Resnet-20 1x | 1,356,098 | 1,143,858 |
> > > > | Sec 5.2 Resnet-20 2x | 5,410,386 |  5,044,756 |
> > > > | Sec 5.2 Resnet-110 1x | 8,620,418 | 7,371,378 |
> > > > | Sec 5.2 Resnet-110 2x | 34,512,276 | 32,014,996 |
> > > > | Sec 5.2 Resnet-20 2x | 5,410,386 | 5,044,756 |
> > > > | Sec 5.4 MLI ResNet | 78,042 | 60,634 |

---

> > > > > ### Comment · Reviewer_ceUs · 2024-08-12
> > > > >
> > > > > I can finally understand more of the purpose of the BNN task. I think you should emphasize in the paper that W-Asym have the same architecture specification and have fewer parameters. Thanks for clarification. I will raise my score.

---

> > > > > > ### Author Response · Authors · 2024-08-12
> > > > > >
> > > > > > Great! We are glad that we could reach consensus on this point.
> > > > > >
> > > > > > We believe we have addressed all of the points brought up in your review. Do you have any other questions or concerns that we could address, which account for the relatively low score that you still give? If so, we can try to address those points in the remaining time of the discussion period.

---

### Official Review · Reviewer_ajZW · 2024-07-14

**Soundness:** 3
**Presentation:** 3
**Contribution:** 4
**Rating:** 7
**Confidence:** 4

**Summary:**

This paper proposes to study the effect of parameter symmetries on the neural networks' training and final properties by analyzing the behavior of networks without such symmetries (or with fewer of them). To do so, the authors develop two methods of parameterizing neural network architectures without parameter symmetries: W-asymmetric parametrization fixes the different subsets of weights in each row of weight matrices to be constant and untrainable, and a $\sigma$-asymmetric network uses a new FiGLU nonlinearity. The paper analyzes the asymmetric properties of both proposed methods theoretically and demonstrates empirically that asymmetry improves the behavior of neural networks in several setups. Specifically, asymmetric neural networks have better linear mode connectivity after training and more stable monotonic linear interpolation between initialization and trained model, are more effective for Bayesian deep learning, and are easier to model with metanetworks.

**Strengths:**

1. The paper proposes a new idea of analyzing the effect of parameter symmetries on neural networks through comparison with asymmetric networks in practical settings. I think this perspective is interesting and has potential for future research.
2. The authors develop easy and effective asymmetric parametrizations and analyze them theoretically.
3. The applicability of the proposed asymmetric perspective is demonstrated on a wide range of related problems.
4. The paper is clearly written and easy to follow.

**Weaknesses:**

My main concerns are related to the limited discussion on the benefits of the proposed perspective in comparison to previous works, asymmetric properties of the proposed methods in practical setups, and poor results of $\sigma$-asymmetric networks:
1. As mentioned in the Related Work section, previous works examined the influence of parameter symmetries on neural network training by changing the optimization schemes instead of the network parametrization. Even though I find the idea of studying the behavior of asymmetric neural networks interesting, it is not clear to me how this perspective is beneficial compared to the constrained optimization one. Adding a more thorough discussion on that in the Related Work section would improve the paper.
2. It is not clear from the paper if W-asymmetric and $\sigma$-asymmetric networks are fully asymmetric or just have fewer symmetries in practical setups. A more accurate discussion on which symmetries are removed by the proposed methods in the experimental setups should be added to the paper. For example, the paper does not cover the normalization layer symmetries, even though layer and batch normalization are used in the experiments. It seems W-asymmetric networks remove normalization symmetries, while $\sigma$-asymmetric ones do not. It may be one of the reasons why $\sigma$-asymmetric networks show weaker results in linear mode connectivity and linear interpolation sections. The ReLU scale symmetry is also not adequately discussed.
3. The analysis of the $\sigma$-asymmetric networks is limited. There is no proof or discussion of the universal approximation for this method. The paper does not explain poor results on linear mode connectivity and linear interpolation and does not include the results on Bayesian neural networks and metanetworks. A more detailed analysis of the $\sigma$-asymmetric networks, or at least a discussion of its suboptimal results, would benefit the paper.

Additionally, I have some minor concerns regarding the experiments:
1. In the Bayesian neural network experiment, the W-asymmetric networks have fewer parameters than standard ones, which may influence the optimal training hyperparameters. Hence, it may be the case that the difference in the results is due to, e.g., different effective learning rates and not the asymmetric network structure.
2. In the metanetwork experiments, standard and W-asymmetric networks have different numbers of parameters. Hence, it may be the case that the difference in the results is due to the different dimensionality of the input space and different optimal training hyperparameters for metanetworks and not the asymmetric network structure.

**Questions:**

I would kindly ask the authors to address the concerns from the Weaknesses section and focus on the following questions:
1. Could you please elaborate on how the asymmetric network perspective differs from the constrained optimization one and in which cases it shows new insights, in your opinion?
2. Could you please clarify whether the W-asymmetric and $\sigma$-asymmetric networks are fully asymmetric in the experiments or not? Which of the known symmetries (neuron permutations, ReLU scaling, pre-normalization parameters scaling, etc.) does each method remove in practice?
3. Could you please comment on the poor results of $\sigma$-asymmetric networks in linear mode connectivity and linear interpolation experiments? Is there any specific reason why the Bayesian neural networks and metanetworks experiments are not conducted for $\sigma$-asymmetric networks?
4. Could you please comment on the technical differences between standard and W-asymmetric networks (optimal hyperparameters and parameter count) in Bayesian neural networks and metanetworks experiments and whether they can affect the conclusions?

Minor questions:
1. Do I understand correctly that the standard deviation $\kappa$ used in W-asymmetric and $\sigma$-asymmetric networks is a hyperparameter tuned separately from the trainable weights? (lines 134 and 163).
2. Do the non-trainable weights of all W-asymmetric networks have the same values in the metanetwork experiment? Does a metanetwork take only the trainable parameters of W-asymmetric networks as input?

**Limitations:**

The authors adequately discuss the limitations of the paper in the appendix.

---

> ### Author Rebuttal · Authors · 2024-08-07
>
> We thank the reviewer for appreciating the novelty of our ideas, the effectiveness of our asymmetric parameterizations, the wide range of problems that we consider, and our writing. We think that we have improved our work through your suggested clarifications and ablations.
>
> > “Even though I find the idea of studying the behavior of asymmetric neural networks interesting, it is not clear to me how this perspective is beneficial compared to the constrained optimization one…”
>
> This is indeed important, see general comment for our elaboration on this.
>
> > “It is not clear from the paper if W-asymmetric and $\sigma$-asymmetric networks are fully asymmetric or just have fewer symmetries in practical setups. A more accurate discussion on which symmetries are removed by the proposed methods in the experimental setups should be added to the paper.”
>
> Good point. We will more clearly explain this. Our theoretical results are summarized in this table:
>
> | --- | W-Asym | $\sigma$-Asym |
> | ---| --- | --- |
> | Permutation | removed | removed |
> | Scale | unclear | removed |
>
> Scale symmetries are removed by FiGLU, as shown in Proposition 2. Although it is not formally proven that W-Asym networks remove scale symmetries, we believe that they do (intuitively, the fixed weights also fix a scale).
>
> > “For example, the paper does not cover the normalization layer symmetries, even though layer and batch normalization are used in the experiments. It seems W-asymmetric networks remove normalization symmetries, while $\sigma$-asymmetric ones do not. It may be one of the reasons why $\sigma$-asymmetric networks show weaker results in linear mode connectivity and linear interpolation sections.  The ReLU scale symmetry is also not adequately discussed.“
>
> This is a good point. We agree that the W-Asymmetric networks appear to remove normalization symmetries, whereas the $\sigma$-Asymmetric ones do not in general.
>
> The ReLU scale symmetry can be handled by changing the nonlinearity to e.g. GELU (Godfrey et al. [19] prove that this does not have scale symmetries). Also, our ResNet experiments still use ReLU nonlinearity, yet they achieve good symmetry breaking.
>
> > “The analysis of the $\sigma$-asymmetric networks is limited. There is no proof or discussion of the universal approximation for this method…”
>
> We did try to prove universal approximation for $\sigma$-Asymmetric networks before submission, but we could not do it for a few reasons; we will add discussion of this to the paper. Classical universal approximation results with MLPs do not apply, because those generally assume elementwise nonlinearities. We do think there is a potential proof via related constructions to the proof of the W-Asym universality and interesting symmetries of SiLU-type nonlinearities [Martinelli et al. 2023], but we leave this to future work.
>
> [Martinelli et al. 2023] Expand-and-Cluster: Parameter Recovery of Neural Networks. https://arxiv.org/abs/2304.12794
>
> > “The analysis of the $\sigma$-asymmetric networks is limited… The paper does not explain poor results on linear mode connectivity and linear interpolation and does not include the results on Bayesian neural networks and metanetworks. A more detailed analysis of the $\sigma$-asymmetric networks, or at least a discussion of its suboptimal results, would benefit the paper.”
>
> Indeed. See our general comment for more on this. We will add more discussion on this interesting point to our paper.
>
> > “In the Bayesian neural network experiment, the W-asymmetric networks have fewer parameters than standard ones, which may influence the optimal training hyperparameters. Hence, it may be the case that the difference in the results is due to, e.g., different effective learning rates and not the asymmetric network structure.”
>
> Good point. We have rerun the experiments, with 90% shallower standard ResNets to match the number of parameters of the W-Asymmetric networks, and we find essentially the same performance: W-Asymmetric test accuracy ($49.3\pm .4$) is still substantially better ($46.8 \pm.9$ for standard and $46.5\pm1.1$ for the shallower one). See our 1-page results PDF for more.
>
> > “In the metanetwork experiments, standard and W-asymmetric networks have different numbers of parameters. Hence, it may be the case that the difference in the results is due to the different dimensionality of the input space and different optimal training hyperparameters for metanetworks and not the asymmetric network structure.”
>
> Again, good point. We have trained a whole new dataset of smaller standard networks, of the same number of parameters as the W-Asymmetric networks (~ 60,000). There is little to no change in the results of these networks and our previous dataset of standard networks, so the difference in results seems to be from the asymmetric network structure. We will include these results in the revision; see our 1-page results PDF for full results.
>
>
> > “Do I understand correctly that the standard deviation $\kappa$ … is a hyperparameter tuned separately…”
>
> Yes, you are correct. We previously said “standard deviation $\kappa$ > 0 that we tune”, but
>  we will note this more clearly.

---

> > ### Comment · Reviewer_ajZW · 2024-08-08
> > **Reviewer's response**
> >
> > Thank you for the detailed response and additional ablations! I find most of my concerns adequately addressed. After reading other reviews and responses, my evaluation of the paper remains very positive. Considering the clarifications on the asymmetric properties of the proposed methods in practice and the low performance of $\sigma$-asymmetric networks, as well as the new ablations, I am raising my score to 7.
> >
> > Generally, I really enjoyed reading this paper =)

---

### Official Review · Reviewer_hbBt · 2024-07-16

**Soundness:** 3
**Presentation:** 4
**Contribution:** 2
**Rating:** 6
**Confidence:** 4

**Summary:**

This paper studies how removing parameter symmetry of neural networks affects the loss landscape, Bayesian Neural Networks and meta-networks. The authors proposed two ways to remove the parameter symmetry: one is similar to pruning that making some parameters untrainable, the other is to adopt non-elementwise activations. Both ways can remove the permutation symmetry in parameter space. After that, the authors empirically demonstrate that removing permutation symmetry substantially 1) make LMC easier to satisfy 2) make BNN's training more efficient 3) improve the performance of meta-network (I am not quite familiar with meta-network or neural functionals.) 4) make training loss along the line segment between initialization and trained parameters more monotonic.

**Strengths:**

1. The writing is clear and easy to follow. Especially, the structure of this paper is clear: front part is about the two ways to remove parameter symmetry, back part is to demonstrate the effect of removing parameter symmetry on loss landscape, BNN, meta-network and MLI.
2. This topic is pretty interesting. Permutation symmetry persists in most neural network architectures and impose structure beyond Euclidean structure to parameter space, however, the community has no deep understanding of how permutation symmetry relates to the success of Deep Learning.
3. The experiments on BNN, meta-networks and MLI are interesting.

**Weaknesses:**

1. A major issue about this paper is that some findings are not new. Especially, some studies have already demonstrated that asymmetric networks are more easily to satisfy LMC. [Cite 1] showed that pruning solutions are within the same basin and pruned networks can be viewed as a special case of W-asymmetric networks.

[Cite 1] Evci, Utku, Yani Ioannou, Cem Keskin, and Yann Dauphin. "Gradient flow in sparse neural networks and how lottery tickets win." In Proceedings of the AAAI conference on artificial intelligence, vol. 36, no. 6, pp. 6577-6586. 2022.

2. Another issue is that in experimental part, the four hypotheses seemingly do not relate to each other. There is not a unified conclusion drawn from the experiments.
3. The most important question about the permutation symmetry remains unresolved. As permutation symmetry holds in most neural network architectures, should we remove permutation symmetry or not? Is the success of deep learning related to the permutation symmetry? This paper seemingly cannot give an insightful answer.
3. In Sec. 5, despite each experiment are motivated from some simple intuition, there is no rigorous theoretical foundation for each hypothesis, which could potentially lower the value of this study.

**Questions:**

N/A

---

> ### Author Rebuttal · Authors · 2024-08-07
>
> We are glad that the reviewer appreciates the writing, topic, and experiments of the paper, and we appreciate the reviewer’s comments. Below, we address them one-by-one.
>
> > “A major issue about this paper is that some findings are not new. Especially, some studies have already demonstrated that asymmetric networks are more easily to satisfy LMC. [Cite 1] showed that pruning solutions are within the same basin and pruned networks can be viewed as a special case of W-asymmetric networks …“
>
> We respectfully disagree. The citation refers to methods for pruning a standard neural net, which are very different from our W-Asymmetric nets. The pruning methods for standard neural networks require specialized training algorithms that differ significantly from standard training (e.g. lottery tickets require repeated training and resetting, dynamic sparse training requires updating connectivity during training).
>
> As noted also by reviewers ceUs and tX9W, our methods are novel. Unliked pruned networks, our W-Asymmetric networks have a fixed sparsity. Thus, they can be trained by standard training algorithms like Adam, just like standard neural networks. This is crucial, since we want our Asymmetric networks to be as similar as standard networks as possible, so that we can gain insights into standard networks. See more on this important point (importance of using standard optimization algorithms) in the general comment.
>
> > “... in experimental part, the four hypotheses seemingly do not relate to each other. There is not a unified conclusion drawn from the experiments.”
>
> This is true, but we do not see this as a downside. In fact, the point of our work is to provide a tool (analysis of asymmetric networks) for us and future works to study the effects of parameter symmetries in many different phenomena at once.
>
> Also, there are some higher-level conclusions that we will emphasize more in the revision. For instance, asymmetric network loss landscapes are somewhat more well-behaved and similar to convex loss landscapes than standard networks, and we show that understanding aspects of neural network optimization requires consideration of parameter symmetries.
>
> > “The most important question about the permutation symmetry remains unresolved. As permutation symmetry holds in most neural network architectures, should we remove permutation symmetry or not? Is the success of deep learning related to the permutation symmetry? This paper seemingly cannot give an insightful answer.”
>
> This is not the point of our work. We probe the effect of parameter symmetries (not just permutation symmetries) in already many different phenomena in deep learning. For instance, such symmetries should be accounted for when merging models, processing models with metanetworks, or training Bayesian neural networks.
>
> > “In Sec. 5, despite each experiment are motivated from some simple intuition, there is no rigorous theoretical foundation for each hypothesis, which could potentially lower the value of this study.“
>
> As mentioned in our abstract, “theoretical analysis of the relationship between parameter space symmetries and these phenomena is difficult.” As in most of the study of deep learning theory, it is very difficult and sometimes intractable to theoretically analyze the effects of symmetries in all of these domains. Even when some theory can be done, it is usually done in very restricted settings (e.g. infinite width, one or two-layer networks, optimization assumed to reach a global optima), which are very unrealistic.
>
> We can consider examples from the literature of theoretical analysis of these hypotheses. In [Ferbach et al. 2024], the authors can only theoretically prove linear mode connectivity up to permutations for the (unrealistic) cases of two-layer mean-field MLPs or untrained MLPs. In [Kurle et al. 2022], parameter symmetries in Bayesian learning are theoretically analyzed for a linear model trained on one datapoint.
>
> The point of our paper is to analyze the effects of symmetries in many domains at once (including some that are not covered in our paper) via empirical studies, which have arguably been more impactful in the study of deep learning. Many other works study deep learning phenomena in a purely empirical way, and they give inspiration for future theory; these include seminal works like those on the Lottery Ticket Hypothesis [Frankle & Carbin 2018], permutation matching for merging [Ainsworth et al. 2022], and early empirical investigations into neural networks [Goodfellow et al. 2014].
>
> **References**
> [Ferbach et al. 2024] Proving linear mode connectivity of neural networks via optimal transport.
> [Kurle et al. 2022] On the detrimental effect of invariances in the likelihood for variational inference.
> [Frankle & Carbin 2018] The Lottery Ticket Hypothesis: Finding Sparse, Trainable Neural Networks.
> [Ainsworth et al. 2022] Git Re-Basin: Merging Models modulo Permutation Symmetries
> [Goodfellow et al. 2014] Qualitatively characterizing neural network optimization problems.

---

> > ### Comment · Reviewer_hbBt · 2024-08-13
> >
> > Thank you for your detailed response and I will maintain my current score.
> >
> > Also, I would like to clarify that
> >
> > > The pruning methods for standard neural networks require specialized training algorithms that differ significantly from standard training (e.g. lottery tickets require repeated training and resetting, dynamic sparse training requires updating connectivity during training)
> >
> > The process of finding the lottery tickets requires repeated training and resetting, however, once the lottery ticket (or subnetwork) is found, the training process of the subnetwork is indifferent from the training of its original network.

---

> > > ### Author Response · Authors · 2024-08-13
> > >
> > > Thank you for the comment. However, we would like to clarify very important differences between lottery tickets and our networks.
> > >
> > > **Lottery tickets have a fixed initialization, W-Asymmetric Networks don't**. Once a pruning mask is found, lottery tickets **must maintain the same fixed initialization** when retrained from scratch. Thus, the only difference between different training runs of a lottery ticket come from things like SGD noise: the two lottery tickets share the same initialization. This is in contrast to our W-Asymmetric networks, which can be trained from any initialization: in our paper's experiments, we always randomly initialize W-Asymmetric networks' learned weights, and never share the initialization of learned weights.
> > >
> > > **Thus, lottery tickets cannot be used for studying many of the phenomena we study with Asymmetric Networks**: metanetworks are often used on networks with different initialization, we cannot use lottery tickets to study standard linear mode connectivity between networks with different initialization, and we cannot use lottery tickets to study monotonic linear interpolation across different initializations. Lottery tickets thus cannot be used for the main purpose of our paper: to explore these diverse phenomena in deep learning from the perspective of parameter symmetries.
> > >
> > > **Moreover, lottery tickets are generally found by training on one task and dataset.** In many regimes, lottery tickets can fail to transfer to other datasets. Morcos et al. 2019 show that lottery tickets can sometimes transfer between datasets for image classification, but there are regimes in which a lottery ticket found one dataset does not do well on other datasets; moreover, we expect this effect to be much worse when changing to different tasks. In contrast, our Asymmetric networks can just be simply initialized, without any special expensive procedure per dataset / task.
> > >
> > > **Our $\sigma$-Asymmetric networks are not similar at all to lottery tickets.** Although the review says "some findings are not new", it does not mention our $\sigma$-Asymmetric networks at all, which are substantially different in structure to pruned networks. This is another valid way of removing some parameter symmetries, which the other reviewers (ceUS, tX9W)
> > >  find is novel: "The methods appear both novel and logical"
> > >
> > > Given these points, we ask the reviewer to reconsider their review. Let us know if you have any further questions or topics for discussion!

---

### Author Rebuttal · Authors · 2024-08-07

We thank the reviewers for their comments and suggestions. We have added new experiments and readied changes for the manuscript, which we think will improve the paper significantly. See our 1-page results PDF for more. Also, we have sent our code for reproducing experiments to the area chair, which should be shared with you too.

Here are our responses to some selected comments by reviewers:

**Why do we not change the optimization algorithm to break symmetries?** (Reviewers hbBt, ajZW)  This is already somewhat touched-upon in our related work section, but we will add more discussion about this:

> “Our models are optimized using standard unconstrained gradient-descent based methods like Adam. Hence, our networks do not require any non-standard optimization algorithms such as manifold optimization or projected gradient descent [5, 54], nor do they require post-training-processing to remove symmetries or special care during analysis of parameters (such as geodesic interpolation in a Riemannian weight space [53]).“

This is a very important point that we will elaborate on further in the revision. We purposefully parameterize Asymmetric networks so that we can use standard optimization algorithms like Adam. This is because the main goal of Asymmetric networks is to provide a counterfactual system that is as similar to standard networks as possible, but with removed parameter symmetries. These other methods that require e.g. optimizing over manifolds like spheres, or iterative retraining for pruning, have significantly different optimization and loss landscape behaviors (e.g. linear interpolation is not even well-defined on general nonlinear parameter manifolds), so they are not suitable for gathering insights into standard networks.

**Masks and constants $\mathbf{F}$ are fixed between runs** (Reviewers ajZW, tX9W). When interpolating between two W-Asymmetric or two $\sigma$-Asymmetric nets, the two networks have the same exact fixed constants $\mathbf{F}$ and masks $M$.

One way to think about this is: when defining a standard network architecture (i.e. a mapping from parameters $\theta$ to functions $f_\theta$), we must specify things like hidden dimension, number of layers, and architecture class (MLP, CNN, etc). When specifying an Asymmetric network of the same architecture class (e.g. a W-Asymmetric MLP), we additionally need to choose masks $M$ and fixed constants $\mathbf{F}$. Thus, since we only ever do linear interpolation between standard networks of the same architecture, we also only ever do linear interpolation between W-Asymmetric networks of the same architecture (so the two networks will have the same masks and constants).

We will include this clarification and way of thinking about the architecture in the rebuttal.

**$\sigma$-Asymmetric network performance** (Reviewers ajZW, tX9W). The $\sigma$-Asymmetric networks do not appear to break symmetries as well as the W-Asymmetric networks, even though both have some theoretical results for symmetry removal.

We have run preliminary empirical tests on many variations of $\sigma$-Asym networks, such as: $sigma(\mathbf{F} sigma(x))$, orthogonal $\mathbf{F}$, sparse $\mathbf{F}$, putting these nonlinearities before the first and after the last layers, adding instead of multiplying the gate, using cosine instead of sigmoid, using square instead of sigmoid, and adding layernorm in the nonlinearity. None of these worked well.

Thus, we feel that there are interesting fundamental questions arising from this relative failure, which could lead to very interesting future work. These are approaches that act on activations, rather than weights, of a neural network. W-Asym Networks act on weights, and do much better in terms of breaking symmetries. We don’t think anything like this has been noted before in the literature (differences between symmetry breaking in weights versus activations).

We will add discussion on these important points in the revised form of our paper.

---

### Decision · Program_Chairs · 2024-09-25

**Decision:**

Accept (poster)

**Comment:**

This paper studies (1) how to construct neural network architectures without "trivial" parameter space symmetries (e.g. permutation or scale invariance) and (2) the downstream, empirical affect of having fewer symmetries on applications ranging from posterior inference to Bayesian NNs to studying loss landscapes.   All reviewers were in favor of acceptance to some degree ( 2 x accept, 1 x weak accept, 1 x borderline accept), applauding the paper's clarity, importance of topic, connection to downstream applications.  The reviewers' most critical issues were about connections to / comparison with optimization-based approached and how the training conditions for the reduced-symmetry networks are substantially different than typical training (jeopardizing the generalizability of the results).  Yet both I and reviewers believe the authors have adequately addressed these concerns in their rebuttal, as the authors' general rebuttal has promised to add more discussion of constrained optimization to the camera-ready version.